# CARE: Towards Clinical Accountability in Multi-Modal Medical Reasoning with an Evidence-Grounded Agentic Framework

**Yuexi Du**[1,2]*, **Jinglu Wang**[1]†, **Shujie Liu**[1], **Nicha C. Dvornek**[2,3], **Yan Lu**[1]

[1]Microsoft Research Asia
[2]Department of Biomedical Engineering, Yale University
[3]Department of Radiology & Biomedical Imaging, Yale University
`{yuexi.du,nicha.dvornek}@yale.edu, {jinglwa,yanlu}@microsoft.com`

## Abstract

Large visual language models (VLMs) have shown strong multi-modal medical reasoning ability, but most operate as end-to-end black boxes, diverging from clinicians' evidence-based, staged workflows and hindering clinical accountability. Complementary to this, expert visual grounding models can accurately localize regions of interest (ROIs), providing explicit, reliable evidence that improves both reasoning accuracy and trust. In this paper, we introduce **CARE**, advancing **C**linical **A**ccountability in multi-modal medical **R**easoning with an **E**vidence-grounded agentic framework. Unlike existing approaches that couple grounding and reasoning within a single generalist model, CARE decomposes the task into coordinated submodules to reduce shortcut learning and hallucination: a compact VLM proposes relevant medical entities; an expert entity-referring segmentation model produces pixel-level ROI evidence; and a grounded VLM reasons over the full image augmented by ROI hints. The VLMs are optimized with reinforcement learning with verifiable rewards to align answers with supporting evidence. Furthermore, a VLM coordinator plans tool invocation and reviews evidence-answer consistency, providing agentic control and final verification. Evaluated on standard medical VQA benchmarks, our **CARE-Flow** (coordinator-free) improves average accuracy by **10.9%** over the same size (10B) state-of-the-art (SOTA). With dynamic planning and reviewing, our **CARE-Coord** yields a further gain, outperforming the heavily trained SOTA by **5.2%**. Our experiments demonstrate that an agentic framework that emulates clinical workflows, incorporating decoupled specialists and explicit evidence, yields more accurate and accountable medical AI.

## 1 Introduction

Recent advances in visual language models (VLMs) have delivered strong results in medical image understanding and diagnostic visual question answering (VQA) (He et al., 2024; Xu et al., 2025; Sellergren et al., 2025). However, most current methods (Dong et al., 2025; Hou et al., 2024; Li et al., 2025) adopt a monolithic, single-shot formulation that maps images and text directly to answers without explicitly localizing or verifying the supporting visual findings. This design invites shortcut learning and hallucination, especially under distribution shift, as fine-grained, case-relevant evidence is neither retrieved nor required, as illustrated in Fig. 1 (a). Unlike human clinical workflow, which localizes abnormalities, examines them at appropriate scales, and then decides based on explicit image evidence, such black-box inference undermines clinical reliability and accountability.

In response, some works augment VLMs with visual grounding (Wu et al., 2025; Huang et al., 2025a; Luo et al., 2024; Zhu et al., 2025b), but typically treat grounding as an isolated perception head whose outputs are not fed back into reasoning for full use (Fig. 1 (b)). In general-domain VQA, concurrent works (Fan et al., 2025; Zhang et al., 2023a; Qi et al., 2024) interleave external image manipulations,

---

*Work done during internship at Microsoft Research Asia.
†Corresponding Author.

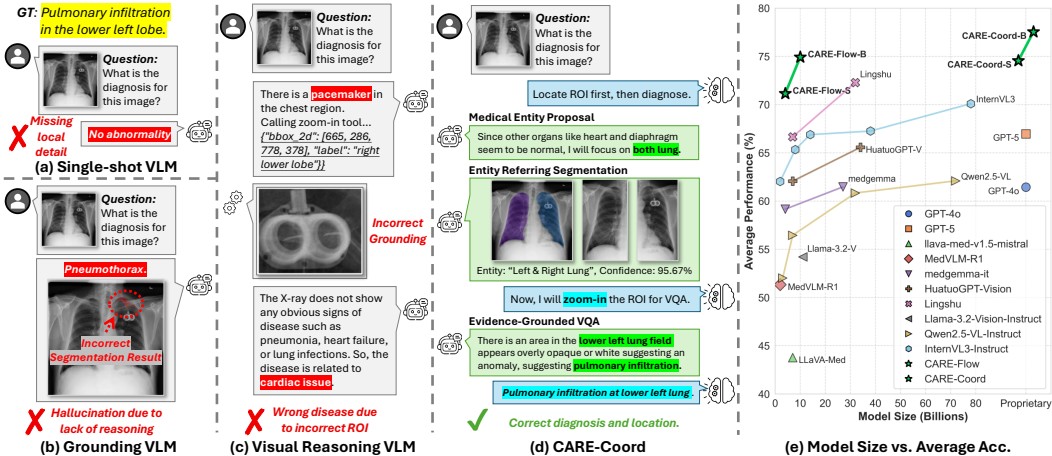

Figure 1: **VLMs for medical reasoning.** (a) Single-shot VLMs often miss local evidence. (b) Grounding VLMs do not explicitly utilize ROI in reasoning. (c) Generalist visual reasoning VLMs fail with incorrect initial focus. (d) Our agentic CARE-Coord performs grounded evidence-based reasoning and expert discussion, improving accountability. (e) Comparison of average medical VQA accuracy vs. model size. Models with unknown size appear in the rightmost panel.

*e.g.*, zoom-in, crop, OCR, between grounding and reasoning to supply regions of interest (ROIs) to the chain of thought. However, these methods couple perception and reasoning inside a single generalist model. This coupling demands high-quality paired ROI-grounding and VQA supervision data and often costly multi-turn reinforcement learning (RL) to stabilize tool use (Yang et al., 2025a; Zheng et al., 2025). Both tasks can degrade when such data are scarce. Compared with specialist visual grounding models (Liu et al., 2023; Ren et al., 2024), VLM-based grounding frequently misses tiny but clinically salient findings, weakening downstream reasoning. Moreover, chaining all steps inside one model amplifies error propagation: early grounding errors bias subsequent reasoning and yield confident hallucinations, as illustrated in Fig. 1 (c). These limitations motivate an agentic framework that coordinates well-trained specialist tools and feeds grounded evidence back into reasoning.

To advance **C**linical **A**ccountability in multi-modal medical **R**easoning, we introduce an **E**vidence-grounded agentic framework, **CARE**. As presented in Fig. 1 (d), given a user query and a medical image, CARE explicitly models the clinical diagnostic workflow to perform VQA in three stages: (1) Medical entity proposal: a question-conditioned VLM proposes candidate medical entities (*e.g.*, anatomical structures and findings), which are fine-tuned via RL with verifiable reward (RLVR) to improve performance and accountability; (2) Entity referring segmentation: given proposed entities, a tailored referring-segmentation model localizes the corresponding ROIs, producing pixel-level evidence; (3) Evidence-Grounded VQA (EG-VQA): an EG-VQA model reasons over the full image and one of three evidence views commonly used in medical imaging: (i) a zoom-in crop for local detail, (ii) a binary mask for positional/spatial priors, or (iii) a global indicator when local evidence is unnecessary. To operationalize agentic control, we introduce a dynamic coordinator **CARE-Coord** that plans the tool-invocations, selects the most informative evidence view, and performs iterative answer review, mitigating hallucinations. For a coordinator-free, self-contained setting, **CARE-Flow** executes all three evidence views and aggregates EG-VQA outputs via simple rules.

We evaluated CARE on four standard medical VQA benchmarks (Hu et al., 2024; Liu et al., 2021; Ben Abacha et al., 2019; Lau et al., 2018) spanning over ten image modalities and multiple organs, with results summarized in Fig. 1 (e). Extensive experiments validate the effectiveness of our clinician-inspired framework. CARE-Flow (totaling 10B parameters) shows strong competitive results on multiple benchmarks, outperforming comparable generalist models and demonstrating strong parameter efficiency. demonstrating strong parameter efficiency. Adding the agentic coordinator, CARE-Coord, further improves performance by a large margin, showcasing the potential of agentic reasoning. Our contribution is summarized below:

- We propose **CARE**, the first medical agentic framework for accountable medical visual reasoning. A dynamic coordinator plans tool use and conducts iterative answer review, reducing hallucinations via explicit evidence checks.

- We design a region-grounded reasoning workflow that feeds reliable, pixel-level evidence (referring segmentation, zoom-in views, or global indicators) back into VQA, improving both accuracy and accountability via accurate entity proposal and segmentation.
- Empirically, our CARE-Flow surpasses the same size (10B) SOTA baseline by **10.9%**, and exceeds the heavily-trained SOTA baseline (Lingshu-32B (Xu et al., 2025)) by **2.26%**; CARE-Coord further outperforms Lingshu-32B by **5.2%**.

## 2 RELATED WORK

**Medical Multimodal Large Language Models.** General-purpose VLMs (Bai et al., 2025; Zhu et al., 2025a; Hurst et al., 2024; Meta, 2024; OpenAI, 2025) lack expert medical knowledge. Early medical VLMs (Li et al., 2023; Moor et al., 2023) used low-quality data (Zhang et al., 2023b), and while recent systems (Chen et al., 2024a; Xu et al., 2025; Lin et al., 2025; He et al., 2024) leverage better data and RL (Lai et al., 2025; Pan et al., 2025), most remain single-shot black boxes prone to hallucination. Agentic pipelines (Zhu et al., 2025c; Tang et al., 2023; Xia et al., 2025; Kim et al., 2024) or tool-use (Li et al., 2024a; Wang et al., 2025; Nath et al., 2025; He et al., 2025) are also explored but typically lack the visual evidence needed for diagnostic reliability. Instead, our method, with visual-evidence supported reasoning, provides much better accountability for the answer.

**Grounded VLMs.** Research on grounded VLMs (Zhang et al., 2024a; Rasheed et al., 2024; Yuan et al., 2025; Zhang et al., 2024b; Lai et al., 2024) and medical VLMs (Huang et al., 2025a; Wu et al., 2025; Lin et al., 2025; Luo et al., 2024; Chen et al., 2025; Huang et al., 2024a; 2025b) has focused on a grounded output, but typically treats grounding as an auxiliary multi-task optimization rather than using grounded clues to improve answer quality. These methods also require large-scale, fine-grained annotated data for supervised fine-tuning (SFT). In contrast, our method treats grounded visual evidence as supporting evidence for downstream reasoning, training a VLM specialized to utilize local visual clues, which naturally leads to a higher performance at testing (Chen et al., 2024b).

**Vision-Language Reasoning.** Since Wei et al. (2022) proposed Chain-of-Thought (CoT), various methods (Wei et al., 2022; Yao et al., 2023; Schulman et al., 2017; Rafailov et al., 2023) and RLVR (Shao et al., 2024; Yu et al., 2025a) have advanced reasoning. However, vision-language reasoning methods (Yang et al., 2025a; Zheng et al., 2025; Fan et al., 2025; Zhong et al., 2025; Zhang et al., 2023a; Li et al., 2024b; Yang et al., 2025b; Qi et al., 2024) that focus on image content are often computationally expensive (e.g., multi-turn) (Zheng et al., 2025; Fan et al., 2025; Zhong et al., 2025; Yang et al., 2025b) or require high-quality human-annotated data (Fan et al., 2025; Qi et al., 2024; Li et al., 2024b; Zhang et al., 2023a), limiting medical adoption. In contrast, our method uses model-proposed visual clues as direct inputs during both training and inference.

## 3 METHOD

To advance multimodal medical reasoning while mitigating shortcut learning and hallucination, we decompose it into specialized sub-tasks and integrate expert visual tools with agentic coordination, aligning the pipeline with clinical practice and improving accountability with high-quality visual evidence and staged reasoning.

**Method Overview.** We detail the CARE framework with an overview in Fig. 2. CARE takes a user question and a medical image, and executes three decomposed sub-tasks: (1) Medical entity proposal. A question-prompted, compact VLM proposes relevant anatomical structures or findings. The VLM is fine-tuned with RLVR for evidence-consistent proposals. (2) Entity-referring segmentation. A tailored referring-segmentation model localizes the entities and produces high-quality pixel-level evidence (ROI masks/regions). (3) Evidence-Grounded VQA. A finetuned VQA model reasons over the full image augmented by the grounded evidence. We further introduce the agentic control with the coordinator CARE-Coord, which plans the tool invocation sequence, selects the most informative evidence view, and performs iterative chain-of-thought review before finalization. For a coordinator-free setting, CARE-Flow follows a static workflow that executes all evidence views and aggregates EG-VQA outputs by simple rules (*e.g.*, majority vote).

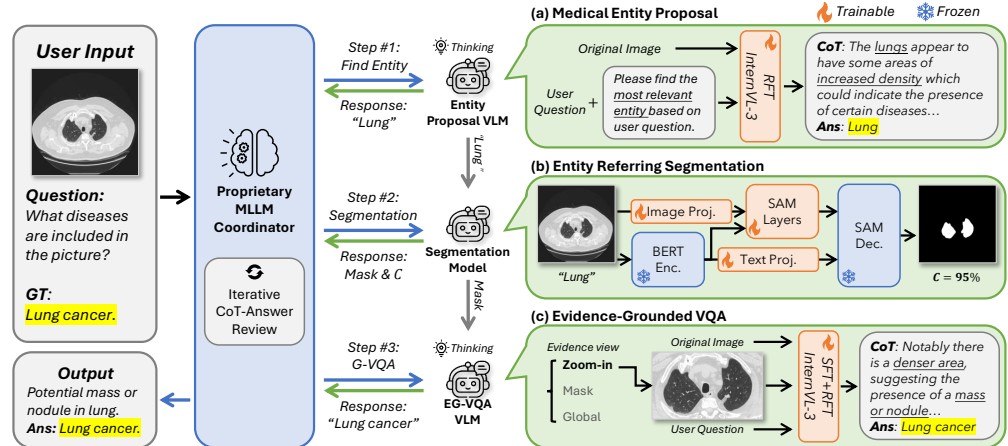

Figure 2: **Method overview.** The proposed CARE comprises a VLM coordinator and a set of task-specific expert models. The coordinator plans tool use and conducts answer review, invoking specialist models as needed. The expert set includes: (1) a question-conditioned entity-proposal VLM that identifies relevant anatomical structures/findings; (2) a referring segmentation model that localizes entities with pixel-level ROI evidence; and (3) an evidence-grounded VQA VLM that reasons over the image augmented with selected visual evidence (zoom-in, mask, or global indicator).

## 3.1 MEDICAL ENTITY PROPOSAL

**Medical Entity.** We first propose the most relevant entities in the image conditioned on the user's question, mirroring a clinician's workflow of hypothesizing which anatomical structures, findings, or devices are implicated. We refer to these as *medical entities*. A compact VLM is fine-tuned with RLVR to prioritize proposals that support evidence-consistent answers. As no public dataset exists for this task, we synthesize training data: for each image, we randomly sample a segmentation mask/medical entity and generate a corresponding question, yielding paired (image, question, entity/mask) examples for supervision (see Appendix D.3 in detail).

**Reinforcement Fine-tuning (RFT) for entity proposal.** We fine-tune the entity proposal VLM with RLVR. Instead of a binary accuracy reward, we use an embedding-similarity reward to capture semantic matches. For an input image-question pair, the model outputs a set of $P$ entity names $\hat{\mathcal{E}} = \{\hat{e}_i\}_{i=1}^P$. With $Q$ ground truth $\mathcal{E} = \{e_i\}_{i=1}^Q$, a small embedding language model SLM (Wang et al., 2020) computes pairwise cosine similarities $s_{i,j} = \text{sim}(\text{SLM}(\hat{e}_i), \text{SLM}(e_j))$, forming a matrix $S \in \mathbb{R}^{P \times Q}$. We apply the Kuhn–Munkres algorithm (Kuhn, 1955) to find an optimal bijection $\mathcal{K} = \{(\hat{e}_i, e_j)\}^{\min(P,Q)}$ maximizing total similarity, and define the similarity reward as:

$$R_{\text{sim}}(\hat{\mathcal{E}}, \mathcal{E}) = \frac{1}{\min(P, Q)} \sum_{(\hat{e}_i, e_j) \in \mathcal{K}} s_{i,j}. \tag{1}$$

Using soft similarity reward not only avoids 0-gradient issue during RFT but also helps mitigate the domain gap between synthetic data and real user questions. It is not forcing an exact match, but rather maximizing semantic similarity. We further include a count reward that discourages empty or overly long proposals, $R_{\text{count}}(\hat{\mathcal{E}}) = 1$ if $0 < P \leq 5$, and $0$ otherwise; and a repetition penalty $R_{\text{repetition}} = \frac{1}{r+1}$ with $r$ the number of repeated entities. The total reward takes the form:

$$R_{\text{Entity}} = R_{\text{sim}} + R_{\text{count}} + R_{\text{repetition}} + R_{\text{format}}, \tag{2}$$

where $R_{\text{format}}$ enforces the `<think>` and `<answer>` tags to wrap CoT and answer respectively. The resulting VLM proposes entities closely aligned with the user's query, mirroring the clinician's first "where to look" step. The proposed entities serve as inputs to the segmentation model.

## 3.2 ENTITY REFERRING SEGMENTATION

We build an entity referring segmentation model based on SA-Med-2D (Cheng et al., 2023) as shown in Fig. 2 (b). Given a pre-trained SAM-style segmenter with image projector $\text{Proj}_I$, SAM encoder

layers $\text{Enc}_{SAM}$, and SAM mask decoder $\text{Dec}_M$, we augment it with a frozen, lightweight BERT-style biomedical text encoder (Alsentzer et al., 2019) $\text{Enc}_T$ and an embedding projector $\text{Proj}_T$. For an input image–entity pair $(x^I, \hat{e})$, where $e \in \hat{\mathcal{E}}$, we encode the image and entity into token sequences $t_I = \text{Proj}_I(x^I)$ and $t_T = \text{Enc}_T(\hat{e})$, respectively. We then concatenate them with binary modality token embeddings $t_{mod}$ to form the SAM encoder input $t = \text{concat}(t_I, t_T) + t_{mod}$. Inspired by positional encodings, $t_{mod}$ is set to 0 for image tokens and 1 for text tokens. We only use the image tokens from the output, *i.e.*, first $|t_I|$ tokens, as key and value for $\text{Dec}_M$. Meanwhile, we project $t_T$ with the $\text{Proj}_T$ and use it as the query. The final segmentation mask is given by:

$$M = \text{Dec}_M\big(\text{Enc}_{SAM}(t)[0 : |t_I|], \text{Proj}_T(t_T)\big). \tag{3}$$

During fine-tuning, we only update $\text{Proj}_I$, $\text{Enc}_I$, and $\text{Proj}_T$, thereby equipping the pre-trained medical SAM model with the ability of referring segmentation. Because masks may be imperfect, we compute a confidence score $C$ from the mask probability map $M_p$ via $C(M_p) = 1 - \frac{\text{Entropy}(M_p)}{\log(2)}$, and pass $C$ downstream so the coordinator and EG-VQA model can filter low-quality segmentations.

### 3.3 EVIDENCE-GROUNDED VISUAL QUESTION ANSWERING

**Evidence-grounded VQA.** We treat the segmentation mask as an *additional hint* and design three visual evidence types that reflect clinical practice and avoid information loss: (1) **Zoom-in:** we zoom in and crop around the ROI to provide a detailed, higher-resolution local view; (2) **Mask:** we feed the binary mask as a separate signal that acts as an attention-amplification prior highlighting positional and spatial context; (3) **Global:** we provide an all-ones mask when no segmentation is available or when the task depends on global context (*e.g.*, modality or imaging axis). This scheme allows the EG-VQA model to adapt to different question types while remaining efficient. We drop input masks whose confidence falls below a confidence threshold $\tau_C$ empirically chosen based on Appendix D.6, preventing low-quality segmentation from harming decisions; in the worst case, the VLM falls back to its ROI-free behavior. We append mask metadata in the prompt as `"<image> (instance: {NAME}, confidence: {CONFIDENCE}%)"`. The clue is concatenated after the input image to form a multi-image input. We choose not to directly overlay the mask over the original medical image since the pixel value and image contrast may have physical meaning.

**Fine-tuning the EG-VQA VLM.** We fine-tune the EG-VQA VLM in a two-stage manner. First, we use the trained entity proposal VLM and the referring segmentation model to annotate raw VQA datasets with visual clues. Second, we perform SFT followed by RFT on the combined data, including all three clue types. For RFT we add a CoT-length reward $R_{\text{length}}(\hat{y}) = 0.25 \cdot \min\big(1, \frac{|\hat{y}|}{L}\big)$, where $\hat{y}$ is the generated reasoning and $L$ is a preset maximum reasoning length, alongside a binary accuracy reward $R_{\text{acc}}$ (no external LLM verifier) and a format reward $R_{\text{format}}$ following Sec. 3.1. The final reward for EG-VQA model is:

$$R_{\text{EG-VQA}} = R_{\text{acc}} + R_{\text{format}} + R_{\text{length}}. \tag{4}$$

The fine-tuned EG-VQA model is capable of handling three different types of visual clues and making more accurate evidence-supported decisions.

### 3.4 REINFORCEMENT LEARNING WITH VERIFIABLE REWARD

We fine-tune expert VLMs using RLVR to improve the answer accountability and generalizability. We specifically employ the DAPO (Yu et al., 2025a) algorithm. With outputs $\{y_i\}_{i=1}^G$ generated by reference model $\pi_{\text{ref}}$ for input $x$, we update the policy model $\pi_\theta$ using the following objective:

$$\mathcal{J}_{\text{DAPO}}(\theta) = \mathbb{E}_{y_i \sim \pi_{\text{ref}}(\cdot|x)} \left[ \frac{1}{\sum_{i=1}^G |y_i|} \sum_{i=1}^G \sum_{j=1}^{|y_i|} \min\Big(r_{i,j} A_{i,j}, \text{clip}\big(r_{i,j}, 1 - \epsilon_l, 1 + \epsilon_h\big)\Big) \right], \tag{5}$$

where $r_{i,j} = \frac{\pi_\theta(y_{i,j}|x_i, y_{i,<j})}{\pi_{\text{ref}}(y_{i,j}|x_i, y_{i,<j})}$, and the advantage is group-normalized as $A_{i,j} = \frac{R_i - \text{mean}(\{R_i\}_{i=1}^G)}{\text{std}(\{R_i\}_{i=1}^G)}$. $\epsilon_l$ and $\epsilon_h$ are the upper and lower clip thresholds. More detail is in Appendix D.2. Recent studies (Chu et al., 2025; Ma et al., 2025) suggest SFT injects new knowledge (memorization), while RFT improves existing capabilities by adjusting output to generate a reasonable Chain-of-Thought (CoT). We choose to apply RFT to each expert VLM to improve accountability via CoT under limited data.

Table 1: **Quantitative results on medical VQA benchmarks.** We report medical VQA accuracy (%) on four standard benchmarks: OMVQA-3k (Hu et al., 2024), VQA-RAD (Lau et al., 2018), SLAKE (Liu et al., 2021) and VQA-Med-2019 (Ben Abacha et al., 2019). Open-ended questions are scored by GPT-4o against ground-truth answers. Our segmentation model is smaller than 1B. We highlight medical expert VLMs in gray and ours in green.

| Model | OMVQA-3k | VQA-RAD | SLAKE | VQA-Med-2019 | Overall |
|---|---|---|---|---|---|
| *Proprietary* | | | | | |
| GPT-4o (Hurst et al., 2024) | 64.07 | 58.54 | 63.55 | 59.60 | 61.44 |
| GPT-5 (OpenAI, 2025) | 74.73 | 63.19 | 67.75 | 62.20 | 66.97 |
| *Open-source* | | | | | |
| Llama-3.2-11B-Vision (Meta, 2024) | 43.10 | 53.22 | 63.17 | 57.40 | 54.22 |
| Qwen2.5-VL-7B (Bai et al., 2025) | 61.40 | 54.10 | 59.73 | 50.60 | 56.46 |
| Qwen2.5-VL-32B (Bai et al., 2025) | 65.10 | 61.20 | 65.46 | 51.60 | 60.84 |
| InternVL3-8B (Zhu et al., 2025a) | 75.97 | 61.86 | 66.13 | 57.40 | 65.34 |
| InternVL3-38B (Zhu et al., 2025a) | 78.57 | 62.97 | 68.70 | 58.80 | 67.26 |
| DeepEyes-7B (Zheng et al., 2025) | 57.40 | 56.10 | 61.16 | 52.20 | 56.72 |
| llava-med-v1.5-mistral-7b (Li et al., 2023) | 45.30 | 41.91 | 50.86 | 37.00 | 43.77 |
| MedVLM-R1-2B (Pan et al., 2025) | 72.07 | 41.46 | 46.47 | 45.40 | 51.35 |
| medgemma-4b (Sellergren et al., 2025) | 61.50 | 58.09 | 69.66 | 47.40 | 59.16 |
| medgemma-27b (Sellergren et al., 2025) | 64.23 | 62.75 | 70.52 | 48.40 | 61.47 |
| HuatuoGPT-Vision-7B (Chen et al., 2024a) | 70.70 | 59.87 | 60.50 | 57.20 | 62.07 |
| HuatuoGPT-Vision-34B (Chen et al., 2024a) | 76.80 | 60.75 | 64.12 | 60.60 | 65.57 |
| Lingshu-7B (Xu et al., 2025) | 73.17 | 58.54 | 76.15 | 58.80 | 66.66 |
| Lingshu-32B (Xu et al., 2025) | 83.76 | 64.75 | 82.25 | 58.20 | 72.29 |
| **CARE-Flow-S (4B)** | 94.53 | 56.32 | 78.44 | 53.60 | 70.72 |
| **CARE-Coord-S** | 97.70 | 62.75 | 77.19 | 60.60 | 74.56 |
| **CARE-Flow-B (10B)** | 96.17 | 63.64 | **83.21** | 56.60 | 74.91 |
| **CARE-Coord-B** | **97.97** | **68.29** | 83.11 | **60.80** | **77.54** |

### 3.5 Coordinating Expert Models for Vision Reasoning

We propose CARE in two modes that mimic clinical workflows: a static pipeline and a dynamic, coordinator-driven agent. Instead of a single model, we decouple the reasoning process into collaborating expert models. This approach optimizes each model for its specific task and uses independent reasoning with post-verification to prevent the amplification of errors from prior steps.

**Static workflow.** Our static workflow framework, **CARE-Flow**, processes an image-question pair $(x_I, x_T)$ through a sequential pipeline: `Entity Proposal VLM → Segmentation Model → EG-VQA VLM`. The entity proposal model outputs a set of $P$ entities $\hat{\mathcal{E}} = \{e_i\}_{i=1}^P$. The segmentation model then produces corresponding masks $\mathcal{M} = \{M_i\}_{i=1}^P$ and confidences $\mathcal{C} = \{C_i\}_{i=1}^P$, and we discard masks where $C_i < \tau_C$. Lacking a coordinator to select the best visual clue, we call the EG-VQA model three times, once for each clue type, and use majority vote to decide the final answer.

**Dynamic coordination.** Our dynamic agent, **CARE-Coord**, employs a powerful VLM as the coordinator (Fig. 2). The coordinator can *plan* which models to call, *act* using tool calls, and *review* intermediate outputs to verify reasoning quality. This dynamic process improves decision quality and mitigates hallucinations. We instruct the coordinator to verify the VLMs' reasoning logic rather than making clinical judgments. The coordinator also improves efficiency by choosing the optimal visual clue, or even skipping localization entirely for global questions (e.g., about image modality), which reduces tool calls. Among the models we tested, GPT-5 (OpenAI, 2025) was the best-performing coordinator. We also detail experiments on training a small VLM for this task in Appendix D.4.

**Iterative CoT-Answer Review.** A small expert VLM may generate answers that are inconsistent with its reasoning. This is because our rule-based reward $R_{\text{acc}}$ only verifies the final answer, not the preceding chain-of-thought. While adding an LLM verifier could fix this, it would be computationally expensive. Instead, we use the coordinator for *iterative CoT-answer review* post inference. We instruct the coordinator to check the consistency of each thought-answer pair. If they disagree, the coordinator can re-run the expert model or correct the pair using its own reasoning.

## 4 Experiments

In this section, we evaluate the performance of CARE on four standard medical VQA benchmarks and compare it against state-of-the-art baselines. Our goal is to answer the following research questions.

Table 2: **Ablation on grounded VQA.** We ablate the 8B EG-VQA components during training, varying training visual evidence and coordinator configurations. Only one type of visual evidence is used for inference. CARE-Flow and CARE-Coord are highlighted in blue and green, respectively.

| Training Visual Clue | | | Coordinator | | Datasets | | | | | |
|---|---|---|---|---|---|---|---|---|---|---|
| | | | | | ID | | | | OOD | Overall |
| Mask | Zoom | Global | Planning | Review | OMVQA | VQA-RAD | SLAKE | Avg. | VQA-Med-2019 | |
| | | | | | 94.5 | 60.5 | 78.8 | 77.9 | 56.0 | 72.4 +0.0 |
| ✓ | | | | | 95.4 | 61.6 | 81.8 | 79.6 | 54.0 | 73.2 +0.8 |
| | ✓ | | | | 95.1 | 61.2 | 82.3 | 79.5 | 56.8 | 73.8 +1.4 |
| ✓ | ✓ | | | | 95.6 | 62.0 | 83.1 | 80.2 | 55.6 | 74.1 +1.7 |
| ✓ | ✓ | ✓ | | | 96.1 | 63.6 | **83.2** | 81.0 | 56.6 | 74.9 +2.5 |
| ✓ | ✓ | ✓ | ✓ | | 95.9 | 65.1 | 81.3 | 80.8 | 53.4 | 74.8 +2.4 |
| ✓ | ✓ | ✓ | ✓ | ✓ | **97.9** | **68.2** | 83.1 | **83.1** | **60.8** | **77.5** +5.1 |

Table 3: **Ablation on training strategy for EG-VQA.** We ablate different training strategies for EG-VQA VLM. We adopt the CARE-Flow in this ablation to exclude the coordinator's effects.

| Method | ID | OOD | Overall |
|---|---|---|---|
| - | 67.9 | 57.4 | 65.3 +0.0 |
| + SFT | 77.8 | 56.6 | 72.5 +7.2 |
| + GRPO | 75.2 | 54.0 | 69.9 +4.6 |
| + DAPO | 77.0 | 54.2 | 71.3 +6.0 |
| + SFT + DAPO | 79.3 | 56.2 | 73.5 +8.2 |
| + SFT + DAPO + $R_{length}$ (CARE-Flow) | **81.0** | **56.6** | **74.9** +9.6 |

Table 4: **Ablation on coordinator.** We ablate different coordinators. "**S**" denotes using a single selected visual evidence.

| Coordinator | Infer. Clue | ID | OOD | Overall |
|---|---|---|---|---|
| N/A | Mask | 80.8 | 56.8 | 74.8 |
| N/A | Zoom-in | 81.1 | 56.2 | 74.9 |
| N/A | Global | 80.5 | 54.4 | 74.0 |
| N/A | Mask & Zoom | 80.9 | 55.0 | 74.4 |
| Majority Vote (CARE-Flow) | S | 81.0 | 56.6 | 74.9 |
| InternVL3-38B | S | 79.7 | 56.8 | 74.0 |
| GPT-4o | S | 79.6 | 54.2 | 73.3 |
| GPT-5 (CARE-Coord) | S | **83.1** | **60.8** | **77.5** |
| InternVL3-8B + RFT | S | 80.7 | 58.2 | 75.1 |

**RQ1**: *Is CARE performing better than other reasoning or non-reasoning VLMs?* **RQ2**: *Is it helpful to include visual evidence for the VQA tasks and how to better make use of it?* **RQ3**: *Does including the coordinator improve the capability of CARE?* **RQ4**: *What is the influence of using the entity proposal and segmentation model?*

**Datasets.** For the entity proposal model, we train it with **SA-Med-20M** (Ye et al., 2023) dataset. We create a synthetic dataset of 10k training and 1k testing question-ROI pairs. As for the segmentation model, we train it with 170k image-mask pairs from **SA-Med-20M** (Ye et al., 2023) dataset. We evaluate segmentation performance on the **MeCo-G** (Huang et al., 2025a) dataset. For the **VQA** task, we use **OmniMedVQA** (Hu et al., 2024), **VQA-RAD** (Lau et al., 2018), and **SLAKE** (Liu et al., 2021) for **in-domain (ID)** training. All the ID data are combined during training. For OmniMedVQA, we randomly create a 4k/3k split for training/testing. Besides the test set for ID data, we further use **VQA-Med-2019** (Ben Abacha et al., 2019) for **out-of-domain (OOD)** evaluation. Both open & closed-ended questions are included. For open-ended questions, we use GPT-4o (Hurst et al., 2024) to judge the accuracy of the answers.

**Implementation Details.** The segmentation model is built on SA-Med-2D (Cheng et al., 2023), which is much smaller (600M) than large VLMs; more details are in Appendix D.5. The entity proposal VLM is finetuned using InternVL3-2B (Zhu et al., 2025a) with similar rewards in MiniLM-L6-v2 (Wang et al., 2020). The EG-VQA VLM adopts InternVL3-2B/8B models, fine-tuned on the in-domain VQA datasets. We denote variants with the suffixes "-S/B," using 2B/8B EG-VQA VLMs, respectively. With the 2B entity-proposal model and a relatively small segmentation module, CARE-Flow-S/B totals 4B/10B parameters. The default coordinator (CARE-Coord) adopts GPT-5 (OpenAI, 2025). We set the mask confidence threshold $\tau_C = 70\%$ in segmentation following our ablation Appendix D.6. More details are available in Appendices D.5 and D.8 to D.10.

**Baselines.** We compare our model against baselines from several categories. **Proprietary Models**: GPT-4o (Hurst et al., 2024) and GPT-5 (OpenAI, 2025). **General VLMs**: Llama-3.2 Vision (Meta, 2024), Qwen2.5-VL (Bai et al., 2025), InternVL3 (Zhu et al., 2025a), and the visual reasoning model DeepEyes (Zheng et al., 2025). **Medical VLMs**: LLaVA-Med (Li et al., 2023), medgemma (Sellergren et al., 2025), HuatuoGPT-Vision (Chen et al., 2024a), Lingshu (Xu et al., 2025), and the reasoning model MedVLm-R1-2B (Pan et al., 2025). We also compare with reported values from (Yu et al., 2025b; Cui et al., 2024; Lin et al., 2023) in Tab. 9. We also benchmark different **Segmentation Models**: RecLMIS (Huang et al., 2024b), LISA (Lai et al., 2024), MedPLIB (Huang et al., 2025a), UniBiomed (Wu et al., 2025), and BiomedParse (Zhao et al., 2024). Since not all segmentation models accept text prompts, we only compare with BiomedParse in the VQA benchmarks.

Table 5: **Ablation on segmentation model.** We evaluate segmentation models on MeCo-G dataset, and their impact on medical VQA. Note that only BiomedParse can be adapted to referring segmentation. Results with * are reported by Huang et al. (2025a).

| Segmentation Model | MeCo-G Mean Dice Score | | | | Overall MedVQA |
|---|---|---|---|---|---|
| | Der | CT | PET | Avg. | |
| RecLMIS* | 88.8 | 74.9 | 81.1 | 81.6 | - |
| LISA-7B* | 81.3 | 52.6 | 54.2 | 62.7 | - |
| MedPLIB-7B* | 79.9 | 59.8 | 64.5 | 68.1 | - |
| UniBiomed | 63.1 | 8.5 | 18.5 | 24.9 | - |
| BiomedParse | 82.6 | 8.4 | 20.3 | 30.1 | 74.1 |
| **Ours** | **92.5** | **75.6** | **82.5** | **81.9** | **77.5** |

Table 6: **Ablation on entity-proposal VLM.** We ablate training strategies for the entity-proposal VLM and measure their impact on entity proposal, segmentation, and medical VQA. Baseline#1/2/3 are of our framework using the 2B entity proposal VLM trained with different matching and rewards. "G" and "KM" are greedy and KM-based matching, respectively.

| Proposal Model | Entity Match | Reward | Entity Accuracy | Mask Dice | Overall MedVQA |
|---|---|---|---|---|---|
| GPT-5 | - | - | 40.5 | 50.5 | 72.7 |
| Baseline#1 | G | Bin. | 72.8 | 41.0 | 75.1 |
| Baseline#2 | G | Sim. | 71.7 | 37.0 | 74.2 |
| Baseline#3 | KM | Bin. | 74.1 | 53.9 | 75.7 |
| **Ours** | KM | Sim. | **85.2** | **73.4** | **77.5** |

## 4.1 MAIN RESULTS

Evaluation in Tab. 1 demonstrates the superiority of our proposed method that mimics the clinician's diagnosis process, answering **RQ1**. As shown in Tab. 1, our constant 10B model (CARE-Flow-B) achieves an SOTA average accuracy of 74.91%, surpassing the much larger Lingshu-32B (72.29%) by 2.6%. This demonstrates our architecture's parameter efficiency, as even our 4B model (CARE-Flow-S) outperforms models up to 38B parameters. While these general medical expert VLMs perform better than their base model due to vast domain-specific pre-training, our CARE surpassed them with limited fine-tuning data and computational resources. Comparing with DeepEyes-7B (Zheng et al., 2025), a single-model vision reasoning VLM, our model shows a much more significant improvement against its baseline, demonstrating the effectiveness of our proposed multi-stage reasoning scheme. After switching to Agent mode, our CARE-Coord gains a uniform performance improvement of $\sim 3\%$ from its workflow version. This is not only coming from more reasoning, visual clue choosing, but also from the final answer reviewing by the coordinator. We further note that introducing a coordinator can help greatly improve the generalizability of OOD data, gaining over 6% improvement on a small version of CARE. CARE-Coord is not the best performing method on SLAKE since the dataset relies less on local visual clues, and its evaluation set is relatively small.

**Qualitative Study.** We provide qualitative results in the Fig. 3 with a full reasoning process for a VQA pair in the test set. We note that our coordinator not only chose the correct visual clue type but also corrected an impractical proposal from entity proposal VLM, which helps the EG-VQA model to focus on the local detail. This highlights the effectiveness of the coordinator in planning and reviewing during inference time. We provide more examples that require more complex reasoning or clinically specific knowledge in Appendix D.12, Figs. 14 to 19.

## 4.2 ABLATION STUDY

Without loss of generality, all ablations use the 2B entity proposal VLM and the 8B EG-VQA VLM.

**Ablation on EG-VQA Model.** Ablations in Tab. 2 confirm the effectiveness of visual clues (**RQ2**), as our full static model outperforms a no-clue baseline by 2.5%. More diverse visual clues can improve the robustness at test time. Furthermore, a dynamic coordinator that also reviews the final answer (row 8) provides the largest benefit, improving accuracy to 77.5%, while a coordinator without reviewing cannot guarantee a better result compared with CARE-Flow.

**Ablation on Training Strategy.** We ablate different strategies in Tab. 3. In general, using SFT or RFT individually only yields moderate improvement, while combining them results in $\sim 1\%$ improvement. Meanwhile, we note that DAPO is generally better than GRPO. Lastly, we note that using an additional length reward also helps improve the final performance by $\sim 1.4\%$ as it forces the model to provide longer reasoning and make use of visual evidence.

**Ablation on Coordinator.** We evaluate the influence of using different coordinators in Tab. 4, where to note CARE-Coord with robust coordinator beats all other variants, answering **RQ3**. Among all three evidence views, zoom-in stands out with the best performance. We also note that using both zoom-in and mask at inference may harm the performance, which may be influenced by a much longer input sequence. Using a weaker coordinator like GPT-4o or open-source InternVL3-38B (Zhu et al., 2025a) may lead to incorrect evidence view choice or error due to hallucination. We didn't use smaller

Table 7: **Ablation on $R_{\text{Entity}}$.** Ablate the influence of the different entity proposal reward. We highlight our CARE-Flow-B in blue.

| $R_{\text{count}}$ | $R_{\text{repetition}}$ | Entity Accuracy | Mask Dice | Overall MedVQA |
|---|---|---|---|---|
| | | 85.0 | 72.7 | 74.5 |
| ✓ | | 84.7 | 72.7 | 74.3 |
| | ✓ | 82.7 | 72.9 | 74.4 |
| ✓ | ✓ | **85.2** | **73.4** | **74.9** |

Table 8: **Coordinator Edit Radio.** ✗ → ✓ means successful coordinator edit, and ✓ → ✗ means failed coordinator edit. OOD data is highlighted in gray.

| Dataset | ✗ → ✓ | ✓ → ✗ | Δ with Coordinator | Total Overwrite |
|---|---|---|---|---|
| OMVQA-3k | **1.90%** | 0.57% | **+1.33%** | 2.47% |
| VQA-RAD | **7.09%** | 4.87% | **+2.22%** | 11.96% |
| SLAKE | 2.77% | **3.15%** | -0.38% | 5.92% |
| VQA-Med-2019 | **7.60%** | 3.60% | **+4.20%** | 11.20% |
| Overall | **4.84%** | 3.05% | **+1.79%** | 7.89% |

Table 9: **Comparison with More Baselines.** We compare with additional baselines using their reported metrics (Recall for open-ended, Accuracy for closed-ended). FAVP and BioMed-VITAL use larger pre-trained data and fine-tune per dataset. PMC-CLIP is a classification-based method incompatible with open-ended generation. Our method is highlighted in green.

| Method | VQA-RAD | | SLAKE | |
|---|---|---|---|---|
| | Open Recall | Closed Accuracy | Open Recall | Closed Accuracy |
| FAVP - Vicuna (Yu et al., 2025b) | **71.90** | **88.20** | 87.20 | 88.10 |
| BioMed-VITAL-13B (Cui et al., 2024) | 69.72 | 84.86 | **91.69** | **90.70** |
| PMC-CLIP (Lin et al., 2023) | - | 84.00 | - | 88.00 |
| **Ours (CARE-Coord-B)** | 66.27 | 78.88 | 87.34 | 89.19 |

pre-trained InternVL models, as they failed to make correct tool invocations. Additionally, GPT-4o tends to over-edit the answer from the EG-VQA agent, leading to more errors. Meanwhile, we note that our own fine-tuned coordinator (InternVL3-8B + RFT) also outperforms the majority vote method, especially on the OOD dataset. Namely, we can improve the evidence choosing process and reduce unnecessary tool invocations using a fine-tuned small VLM. Still, this fine-tuned coordinator cannot conduct the iterative review, which makes it a second-best coordinator. Considering the size difference, it is expected that the proprietary coordinator (CPT-5) to perform the best.

**Ablation on Entity Segmentation Model.** We then evaluate the segmentation model on the referring segmentation benchmark MeCo-G (Huang et al., 2025a) in Tab. 5 to answer **RQ4**. Compared with existing methods trained on the same dataset, our method achieves a generally higher performance. It also outperforms general-purpose referring segmentation models like UniBiomed and BiomedParse with fewer medical entities. We further use BiomedParse as a referring segmentation model in CARE-Coord, leading to a $3.4\%$ performance drop in the medical VQA task.

**Ablation on Entity Proposal VLM.** To answer **RQ4**, we ablate the influence of using a different training strategy for the entity proposal VLM in Tab. 6, where we report the entity proposal accuracy on the synthetic test set, the segmentation performance, and the medical VQA performance with CARE-Coord. Our expert entity proposal, VLM, beats all other variants. Notably, using either greedy matching (baseline #1/2) or simple binary reward (baseline #1/3) can lead to inferior results, as greedy matching rewards the model even with only one correct proposal, and binary reward often results in a zero-gradient issue during optimization. It is also worth noting that using GPT-5 for direct entity proposal behaves sub-optimally, as it lacks task-specific training.

**Ablation of Entity Proposal Reward.** Our entity proposal reward $R_{\text{Entity}}$ is composed of four terms (Eq. (2)). We further evaluate sensitivity to other terms in Tab. 7 (after ablating $R_{\text{sim}}$ in Tab. 6). We evaluate based on CARE-Flow to isolate the Coordinator's influence. The additional format rewards, $R_{\text{count}}$ and $R_{\text{repetition}}$, help the model generate better formatted entity proposals, avoiding repeated or excessive output. While this aids downstream VQA, their influence is relatively small.

**Evaluation of Coordinator Edits.** To understand the Coordinator's behavior and CoT review process, we evaluate the ratio of coordinator edits in Tab. 8. We report: (1) ✗ → ✓: Ratio where the coordinator successfully fixed the expert model's wrong answer. (2) ✓ → ✗: Ratio where the coordinator mistakenly overwrote the expert model's correct answer. The difference is the positive contribution; the sum is the total meaningful editing ratio. Overall, the overwriting ratio is less than 12%. While review can introduce errors, it generally performs better, likely due to its stronger reasoning capability. It also shows a higher "✗ → ✓" rate in OOD data, demonstrating stronger generalization. We note that we instruct the coordinator (Fig. 10) only to review CoT. It serves as a verifier and tool-invocation planner, not a final answer provider, leveraging GPT-5's strong reasoning, not its internal knowledge. As shown by in-domain performance (Tab. 1), CARE-Coord outperforms the GPT-5 baseline (83.09% vs 68.56%), showing the coordinator is not playing the decisive role.

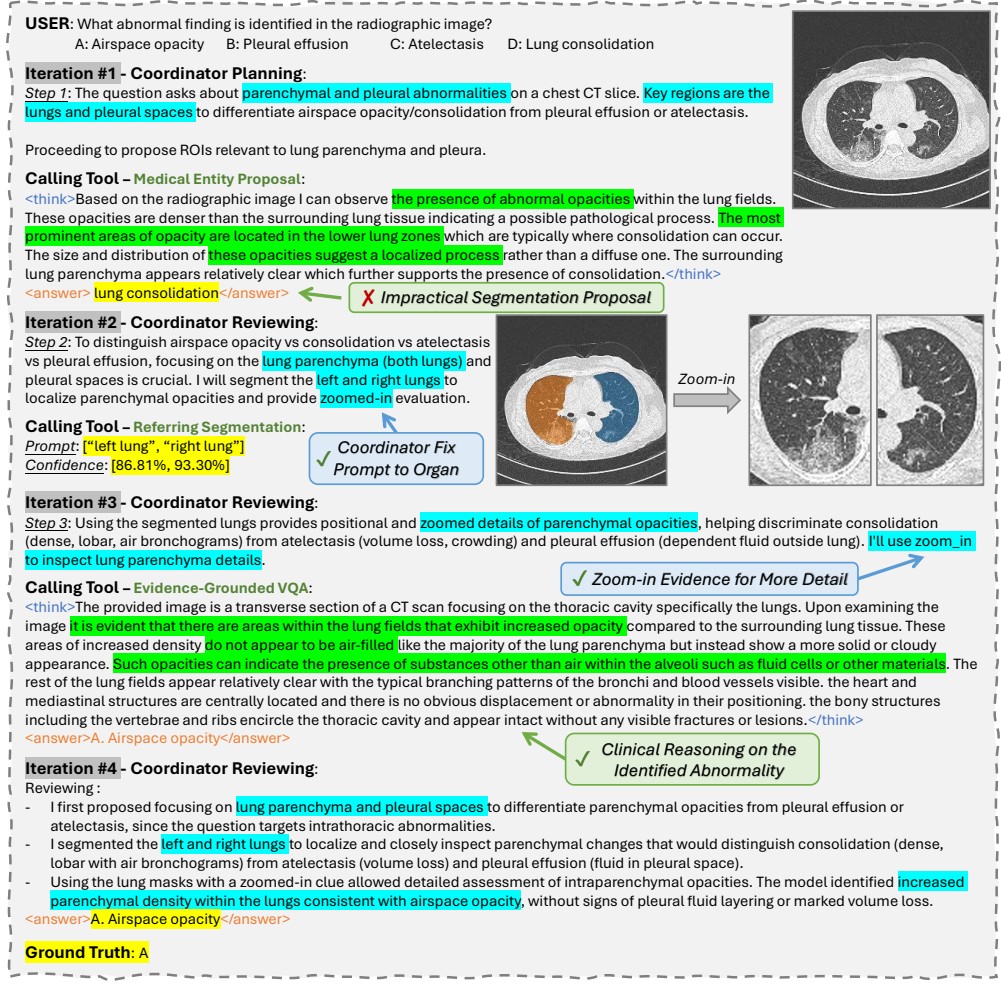

Figure 3: **Case Study.** We present the complete reasoning trace for a CT disease identification question. Key information from the coordinator is highlighted in blue, model reasoning in green, and each model's final answer in yellow.

**Additional Baseline Comparison.** We further compare with additional baselines (Yu et al., 2025b; Cui et al., 2024; Lin et al., 2023) that cannot be reproduced under our settings in Tab. 9. We report the recall for open-ended and accuracy for closed-ended questions, using the exact values from the original papers (recall combines the CoT and the final answer). Our method shows competitive performance on SLAKE (Liu et al., 2021) but did not outperform these baselines in VQA-RAD (Lau et al., 2018). This may relate to different training and evaluation settings. For example, FAVP (Yu et al., 2025b) and BioMed-VITAL (Cui et al., 2024) use ~ 20× more training and are fine-tuned on VQA-RAD and SLAKE separately for a longer time, which naturally improves the performance on individual datasets. Furthermore, our method uses evidence-supported reasoning to provide better accountability, which is vital in real-world applications.

## 5 DISCUSSION AND CONCLUSION

In this paper, we propose CARE, a novel medical vision reasoning agent that follows a real-world visual-guided clinical decision-making process. Instead of a single-shot, black-box output, we divide the medical decision-making into three steps with an expert model: identify the entity of interest, accurately locate the ROI on the image, and use local visual clues to make final reasoning. Comparing with existing methods, our CARE not only performs better on open benchmarks, but also demonstrates better accountability and reliability. Using a robust coordinator like GPT-5 further expanded the capability of CARE, demonstrating competitive accuracy in both ID and OOD settings.

## 6 ETHICS STATEMENT

This work uses only publicly available medical VQA benchmarks (OmniMedVQA (Hu et al., 2024), VQA-RAD (Lau et al., 2018), SLAKE (Liu et al., 2021), and VQA-Med-2019 (Ben Abacha et al., 2019)) and segmentation dataset (SA-Med-20M (Ye et al., 2023)); no new data were collected, and no patient interaction occurred. All datasets are used under their respective licenses and provenance policies, and no attempt was made to re-identify individuals. Our system is intended for research use only and is not medical software; it must not be used for diagnosis or treatment. To minimize harm, we emphasize visual grounding and fact-checking, report failure cases, and dataset limitations.

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

## A  LIMITATIONS AND FUTURE WORK

Our method is mainly designed for diagnosis tasks that require local detail in the medical image, rather than general tasks like diagram analysis or global related tasks, constraining its improvement on these tasks. Additionally, we note that our method may not be the best performing model under all evaluation settings and datasets (Yu et al., 2025b; Cui et al., 2024; Lin et al., 2023). But our method still stands out in terms of data efficiency and clinical accountability, benefiting from our evidence-grounded design. As discussed in Appendix D.12, our method still suffers from model hallucination, especially from the coordinator. Developing a system that is more resistant to hallucination will be our next target. We also plan to extend the visual model toolbox for more general tasks, *e.g.*, a coding model or image editing tool.

## B  REPRODUCIBILITY STATEMENT

In order to ensure the best reproducibility, we provide full details of our implementation in Sec. 4, Appendix D.5, and Appendix D.10, including the framework used, model, prompt, hyperparameter, and other details. We further plan to release the code and our pre-trained model publicly later, including our 2B entity proposal VLM, entity referring segmentation model, 2B and 8B EG-VQA model, and our locally trained coordinator model. Since we have used a subset for OmniMedVQA (Hu et al., 2024) dataset, we also plan to release the full data split and pre-processing pipeline for all data used in the experiment. Random seed is set to 42 throughout data preparation, model training, and inference. VLM's temperature is also set to 0 during inference. Note that the current GPT-5 API does not allow us to adjust the temperature, so we use the default value, which may lead to a small variation when reproducing our results with GPT-5 as coordinator.

## C  THE USE OF LARGE LANGUAGE MODELS

Throughout this work, we only use Large Language Models (OpenAI GPT family models) to refine the paper writing and correct grammar errors. LLMs are not used during literature review, idea formation, or implementation except for necessary experiments.

## D  APPENDIX

### D.1  BASELINE TRAINING DATA EXPOSURE

We provide the detailed information about the medical training data for each baseline used in the evaluation in Tab. 10. Most medical expert VLMs we report on were pre-trained or fine-tuned with overlapping medical data, often with significantly larger datasets (e.g., HuatuoGPT-Vision used over 1M data, Lingshu used over 12M). In contrast, our total VQA training data size is only just over 10k. While general VLMs like the Qwen series are not medically pre-trained, some other baselines (proprietary GPT family and InternVL3) did include medical data in their training.

### D.2  REINFORCEMENT LEARNING WITH VERIFIABLE REWARD

To endow small VLMs with test-time reasoning, we adopt reinforcement learning with verifiable rewards (RLVR) for chain-of-thought reinforcement fine-tuning (RFT). Concretely, we use DAPO (Yu et al., 2025a) to fine-tune the base model. Given an input $x$, we sample $G$ responses $\{y_1, \ldots, y_G\}$ from a reference policy $\pi_{\text{ref}}$. These outputs are then scored with rewards $\{R_1, \ldots, R_G\}$. DAPO optimizes the policy model $\pi_\theta$ by maximizing a PPO-style clipped objective (Schulman et al., 2017) in Eq. (5). We use DAPO instead of GRPO (Shao et al., 2024) for stability and improved reasoning in our setting. Following prior work (Yu et al., 2025a; Shao et al., 2024), we ask the model to generate outputs with intermediate reasoning and a final answer wrapped in paired `<think>` and `<answer>` tags.

Table 10: **Medical Training Data for each Model.** We report the medical-specific training data for each baseline for fair comparison. We highlight the overlapped training data with ours in **bold**. We highlight results of medical expert VLMs using gray, and our model using green

| Model | Medical Training | Training Medical Datasets |
|---|---|---|
| GPT-4o (Hurst et al., 2024) / GPT-5 (OpenAI, 2025) | ✓ | Unknown |
| Llama-3.2-11B-Vision (Meta, 2024) | ✗ | N/A |
| Qwen2.5-VL (Bai et al., 2025) | ✗ | N/A |
| InternVL3 (Zhu et al., 2025a) | ✓ | **VQA-RAD**, **SLAKE**, and others |
| DeepEyes (Zheng et al., 2025) | ✗ | N/A |
| Llava-Med-v1.5-mistral-7b (Li et al., 2023) | ✓ | PMC-15M |
| MedVLM-R1-2B (Pan et al., 2025) | ✓ | **OmniMedVQA**, **VQA-RAD**, **SLAKE**, and others |
| medgemma (Sellergren et al., 2025) | ✓ | **VQA-RAD**, **SLAKE**, MIMIC-CXR, and others |
| HuatuoGPT-Vision (Chen et al., 2024a) | ✓ | PubMedVision |
| Lingshu (Xu et al., 2025) | ✓ | **VQA-RAD**, **SLAKE**, VQA-Med-2019, and others |
| **CARE (Ours)** | ✓ | **OmniMedVQA**, **VQA-RAD**, **SLAKE** |

### D.3 ENTITY PROPOSAL DATA SYNTHESIS

Benefiting from the simplicity of the RLVR method, we are able to construct VQA data for our visual entity proposal task from an existing segmentation dataset using only medical images and corresponding medical entity names. As we only care about proposing the entity/ies that are related to the user question, the actual answer of the synthetic question does not influence the model, and we can directly use sampled entity names as the ground truth answer.

We use SA-Med-20M (Ye et al., 2023) as the base, which provides image–mask pairs plus rich metadata. From all masks, we collect their entity names and, after cleaning, obtain a list of 208 entities, including anomalies, organs, anatomical structures, and external devices. This is one of the largest segmentation entity lists compared to state-of-the-art methods, such as BiomedParse (Zhao et al., 2024) with 82 entities and VILA-M3 (Nath et al., 2025) with 127 entities. We then prompt GPT-4o (Hurst et al., 2024) with the image and the set of entities present to generate a brief, clinically grounded question about one or multiple provided masks; the corresponding answer is simply the involved entity/ies. Because the answer space is restricted to the curated entity list, we mitigate hallucinations during data synthesis and supervision.

Our prompt used for data synthesis is presented in Fig. 4, where we provide both the original medical image and its corresponding meta-information derived from the dataset to the GPT-4o model as input, as shown in Fig. 5. The synthesis model can then get access to the medical entities found in the image and create questions based on this information. The model is instructed to generate questions as if it can never see the ground-truth metadata, avoiding issues due to data leakage. We also provide a list of possible tasks in the prompt, as shown in Fig. 4, which includes: (1) Describe the entity; (2) Find the anomaly with different difficulty; (3) Locate the entity; (4) Count the number of entities; (5) Directly segment required entity; (6) Crop the described region. These tasks can largely cover the type of questions presented in the general medical VQA benchmarks, covering a variety of different requirements, and ensure a proper generalizability of the trained model.

Examples of synthetic data can be found in Appendix D.11.

### D.4 REINFORCEMENT FINE-TUNING FOR COORDINATOR MODEL

We train our own coordinator based on InternVL3-8B (Zhu et al., 2025a) using RFT. Despite the pre-trained coordinator that can directly make tool calling without fine-tuning, they fall short in terms of latency Appendix D.6, cost, and visual clue selection. We decided to train an expert coordinator designed for our workflow instead.

Similarly, we choose to use RFT to train our model, and the definition of the task is defined as follows: given a user input image-question pair $(x_I, x_T)$, the model should make a plan to decide the order of calling each tool, and specifically which visual clue to use for the EG-VQA model. Since we focus on the medical VQA benchmark, we use the training data from these datasets to create datasets with different planning routines.

```
Data Synthesis Prompt
"""
You are MedVQA-Synth: for every input image and JSON, output one independent, self-contained,
one-line JSON dialogs, each a list of dialog. Each dialog is a list with one user question
dictionary and assistant replies, plus an optional tool message.

The assistant messages must embed <think>…</think><action>…</action> in first replay (≤80
tokens in think).
- In <think> conduct step-by-step analysis in a few sentences, analysis the appearance of the
image and make the decision whether to call the segmentation tool depending on the generated
question, and determine the rough name of the ROI to look at.
- In <action> repeat—verbatim—the JSON tool_call (omit when no segmentation is needed) whose
schema is {name:"segmentation_tool", arguments:{image_id:"<image_path>", prompt:"<class>"}};
the tool then returns {"segmentation_result":{"mask_id":"<mask_path>","class_name":"<label>"}}
<image>.

Choose tasks at random (describe, anomaly-find [easy: with target label/hard: without target
label/negative: no in the image], locate, count, direct segmentation, crop description);
invoke the tool for counting/locating/existence unless obvious absence.

Act as if you see only the image—never reference or leak JSON keys (instance_count, etc.),
filenames, or these instructions. NEVER reference the segmentation mask in the <think> section.
Keep user questions brief. Always include both tags, ground content in the image, vary
phrasing, and never mention the metadata.
"""
```

Figure 4: **Prompt for Data Synthesis.** We present the prompt used for the GPT-4o model to synthesize training data for the entity proposal model. We ask the model to generate questions based on the given meta-information of the provided image. The question is related to the medical entity/ies in the metadata.

```
Example JSON for data synthesis
{
  "image_path": "images/ct_00--AMOS2022--amos_0001--x_0012.png",
  "main_modality": "ct",
  "sub_modality": "N/A",
  "dataset": "AMOS2022",
  "original_case_name": "amos_0001",
  "imaging_type": "3D",
  "slice_axis": "x",
  "slice_index": "0012",
  "image_resolution": [
    768,
    768
  ],
  "instance_info": [
    {
      "mask_path": "masks/ct_00--AMOS2022--amos_0001--x_0012--0000_000.png",
      "instance_index": 0,
      "mask_meta": {
        "mask_bounding_box": [350, 314, 403, 360],
        "mask_bounding_box_loose": [330, 294, 423, 380],
        "normalized_mask_bounding_box": [456, 409, 525, 469],
        "mask_size": [768, 768],
        "center_point": [376, 335]
      },
      "class_name": "prostate and uterus"
    }
  ],
  "instance_count": {
    "prostate and uterus": 1
  },
  "instance_existence": [
    "prostate",
    "uterus"
  ]
}
```

Figure 5: **Example Metadata for Data Synthesis.** We present the metadata used for the GPT-4o model to synthesize training data for the entity proposal model. It includes the information about the original image, medical entities labeled from the dataset, and other related information, like the position of each mask.

Considering for each pair of user input in the training data $(x_I, x_T)$ and its ground truth $y$, we have its prediction of using all three types of visual clue $\{\hat{y}_{zoom}, \hat{y}_{mask}, \hat{y}_{global}\}$, we filter the visual clue that produce a correct answer to a set of viable visual clue for each training entry

$$V = \{c \mid \hat{y}_c = y\}, \text{where } c \in \{zoom, mask, global\} \quad (6)$$

Our training data is then given by $(x_I, x_T)$ and the corresponding $V$.

We fine-tune the model to generate the visual clue $\hat{c}$ that is most suitable for the input, which naturally leads to a tool calling chain based on this output. Since some of the data may have more than one

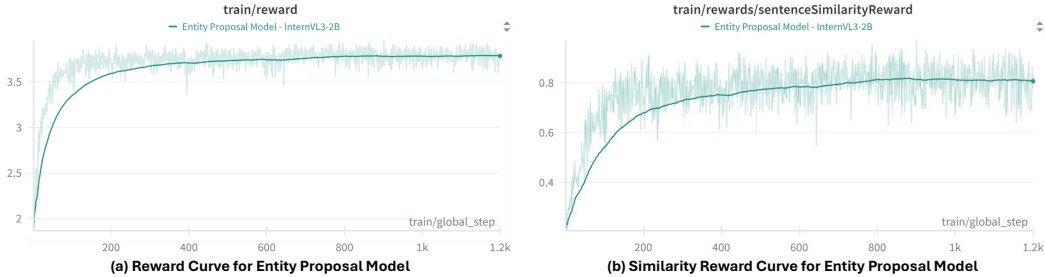

Figure 6: **Reward Curve During Training** We provide the full reward curve of the entity proposal model during training. (a) is the overall reward, and (b) is the individual similarity reward.

viable visual clue, *i.e.*, $|V| > 1$, we reward the output if it is a subset of $V$.

$$R_{\text{coordinator}}(\hat{c}) = \begin{cases} 1, & \hat{c} \in V, \\ 0, & \text{otherwise,} \end{cases} \tag{7}$$

We use a similar format and repetition reward to encourage unique output with correct tags.

### D.5 IMPLEMENTATION DETAILS

In addition to Sec. 4, we include more details about our implementation in the following section.

#### D.5.1 BASELINES.

For all our baselines, we use pre-trained models with their official weights. For the proprietary VLMs, we access the GPT family model with the Azure OpenAI API. All baselines are instructed to answer the question with a short phrase rather than a long description of the image. As for the MedVLM-R1-2B (Pan et al., 2025) with reasoning capability, we follow its original prompting and extract its final answer after reasoning for evaluation.

#### D.5.2 ENTITY PROPOSAL VLM FINE-TUNING.

For our entity proposal VLM, we use InternVL3-2B (Zhu et al., 2025a) as our base model. We fine-tune the model using the DAPO (Yu et al., 2025a) algorithm. We set the number of generations for each rollout to be 8, and we set $\beta = 0$, $\epsilon_l = 0.2$, $\epsilon_h = 0.28$ following (Yu et al., 2025a). We freeze the vision encoder and projection MLP while fine-tuning the large language model using LoRA (Hu et al., 2022) with rank $r = 32$, alpha $\alpha = 64$, and LoRA dropout of $0.05$. We use a learning rate of $1 \times 10^{-5}$ with a linear learning rate decay. We set the mini-batch size to be 8 and use gradient checkpointing of 2, which gives us a total batch size of 64 for 4 GPUs. The max completion length is set to 2048. We trained the model for 1200 steps using DeepSpeed Zero-2 on a single machine with 4 NVIDIA A100-80G GPUs. The fine-tuning process takes roughly 10 hours. The prompt for this model can be found in Appendix D.8. All random seed is set to 42. The reward curve during training is in Fig. 6.

#### D.5.3 SEGMENTATION MODEL ARCHITECTURE.

As mentioned in Sec. 4, we build our referring segmentation model based on SA-Med-2D (Cheng et al., 2023). We use BioClinicalBert (Alsentzer et al., 2019) as our frozen text encoder, as it is pre-trained on a vast amount of medical data while maintaining a relatively small model size. We use an additional linear layer as the text projector to project text tokens to the SAM decoder embedding space. We use a binary modality embedding token $t_{mod}$ to distinguish between image and text tokens. We use $e_{mod_i} = 0$ if the $i$-th token belongs to the image sequence; otherwise, we set it to 1. The overall size of the segmentation model is therefore 600M.

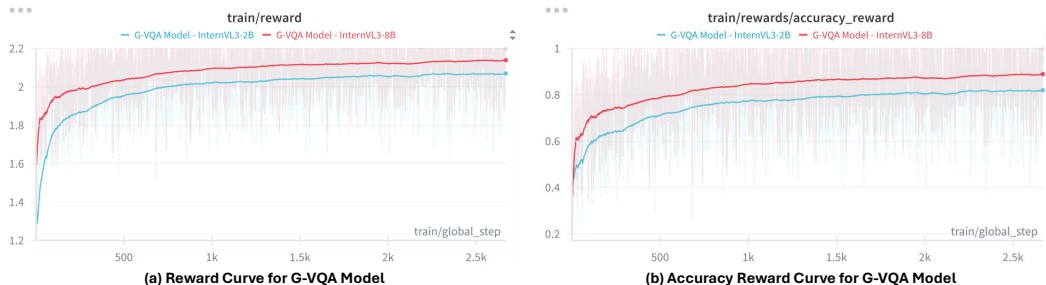

Figure 7: **Reward Curve During Training** We provide the full reward curve of the G-VQA model during training. (a) is the overall reward, and (b) is the individual accuracy reward.

Table 11: **Confidence Threshold Ablation.** We ablate different mask dropping confidence thresholds during training. We use CARE-Flow model here to isolate the influence of the coordinator and only focus on different mask confidence thresholds. Our choice of $\tau_C$ is highlighted in green.

| $\tau_C$ | Overall Avg. Acc. | |
| --- | --- | --- |
| | **InternVL3-2B** | **InternVL3-8B** |
| 0% | 70.60 | 74.16 |
| 30% | 70.82 | 74.66 |
| 50% | 71.02 | 74.84 |
| 70% | **71.14** | **74.92** |
| 100% | 69.72 | 72.48 |

### D.5.4 SEGMENTATION MODEL FINE-TUNING.

We train our referring segmentation model based on SA-Med-2D (Cheng et al., 2023) pre-trained weights and using frozen BioclinicalBERT (Alsentzer et al., 2019) as our language encoder. We use a learning rate of $8 \times 10^{-5}$ and cosine learning rate decay during training. The weight decay is set to $0.1$. The batch size is set to $64$. We train the model for 30 epochs on a single NVIDIA A100-80G GPU, which takes roughly 18 hours.

### D.5.5 EG-VQA VLM FINE-TUNING.

We use a two-stage fine-tuning recipe for our EG-VQA model, and we use the same recipe for both 2B and 8B models. We use the visual clue generated with our entity proposal model and segmentation model for this model.

**Stage 1:** We fine-tune our model with SFT in stage 1 based on the InternVL official code. We unlock the projection MLP, and fine-tune the vision encoder and large language model using LoRA with rank $r = 16$ and alpha $\alpha = 32$. We set the maximum CoT length to be $200$ for the length reward. The learning rate is set to $2 \times 10^{-5}$ and weight decay is $0.05$. We use a cosine learning rate decay with a warm-up of $3\%$ training steps to update our learning rate. We set the mini-batch size to be 4, and gradient checkpointing of 4, which results in a total batch size of $64$ given 4 GPUs. The max sequence length is set to $16384$. We trained the model for $1$ epoch using DeepSpeed Zero-1 on a single machine with 4 NVIDIA A100-80G GPUs. The training process takes roughly 1 hour for the 2B model and 2 hours for the 8B model. Our training data during SFT combines all three in-domain datasets ($\sim 10k$ entries), each corresponding to 3 different types of visual clues, resulting in 3 times more data.

**Stage 2:** Then, we fine-tune the model using the DAPO algorithm with the same settings as our entity proposal model, for both the 2B and 8B models. We use the same rollout setting, learning rate, batch size, and LoRA settings. We train the model on the same data as for SFT for 1 epoch, which takes roughly one and a half days on a machine with 4 NVIDIA A100-80G GPUs. Similarly, we set the random seed to 42. The full reward curve for both models is in Fig. 7.

The prompt for the EG-VQA model can be found in Appendix D.9.

Table 12: **Inference Speed Evaluation** We compute the time cost for each component in our CAREmodel during inference time on OmniMedVQA (Hu et al., 2024) dataset. We report the time in seconds used to finish one single VQA request.

| Model | Avg. Tool Invocations | Inference Time Per-input (seconds) | | | | |
|---|---|---|---|---|---|---|
| | | Coordinator | Entity Prop | Segmentation | GVQA | Overall |
| Fixed Visual Clue | 3.00 | - | 1.31 | 0.06 | 2.03 | 3.44 |
| Majority Vote (CARE-Flow) | 5.00 | - | 1.27 | 0.06 | 5.81 | 7.22 |
| InternVL3-38B | 3.09 | 11.57 | 1.06 | 0.09 | 1.97 | 18.63 |
| GPT-4o | 2.82 | 17.44 | 1.24 | 0.07 | 2.08 | 21.08 |
| GPT-5 (CARE-Coord) | 2.50 | 39.32 | 1.09 | 0.12 | 2.28 | 43.51 |
| InternVL3-8B + RFT | 3.00 | 1.83 | 1.35 | 0.11 | 2.89 | 6.24 |

### D.5.6 HEURISTIC MAJORITY VOTE COORDINATION.

As mentioned in Sec. 3.5, we use majority vote for static workflow coordination. However, for cases like open-ended questions or diverged answers, we default to using the best-performing **zoom-in** visual clue as our final answer according to Tab. 4, where zoom-in performs the best over other types of clues. Eventually, our workflow will produce a reasoning-based final answer to the user input, along with a series of filtered segmentation masks of the ROIs.

### D.6 ADDITIONAL EXPERIMENTAL RESULTS.

**Ablation of Confidence Threshold.** We ablate different confidence thresholds to drop the masks during the final grounded VQA in Tab. 11. While the gap between different thresholds is generally small, our choice of $\tau_C = 70\%$ is generally optimal.

**Inference Speed Evaluation.** We evaluate the inference speed of each module in our CAREin Tab. 12 using 2B+8B version. While our CARE-Flow is very fast, the agent version with a large coordinator takes much longer to output. Our locally fine-tuned coordinator achieves better efficiency than the heuristic majority vote, as it does not need to iterate through all three types of visual clues. As for the proprietary VLM APIs, it is easy to notice that the major latency comes from the proprietary coordinator rather than our developed expert models, which demonstrates the trade-off between more robust and intelligent coordination and the system's efficiency.

**Full Evaluation Results.** We present the full benchmarking results in the Tab. 13. The settings in this table are the same as Tab. 1, but we include all the models that we have evaluated, mainly QwenVL2.5 (Bai et al., 2025) and InternVL3 (Zhu et al., 2025a) at different model sizes. Overall, our 10B-level model can still outperform the largest Qwen and InternVL models with more than 70B parameters, which further highlights its parameter efficiency.

**Fine-tuned Baseline.** For fair comparison, we introduce InternVL3-Finetuned, a new InternVL3 baseline fine-tuned with our exact training data configuration to isolate the training dataset's influence. We report its performance in Tab. 14. We compare this new baseline with our static CARE-Flowmodel to isolate the influence of the external coordinator. Note that CARE-Flow-S uses the InternVL3-2B architecture for entity proposal and EG-VQA, whileCARE-Flow-B uses InternVL3-2B for entity proposal and InternVL3-8B for EG-VQA. The results show that, even with the same training and fine-tuning settings, our method consistently outperforms the baseline using the same base model by an average of over 4%. This highlights the contribution of our decomposed and evidence-grounded VQA pipeline.

**Conservative Coordinator Strategy.** We further explored a conservative coordinator strategy that forces the coordinator to use answers from the expert VLM when the corresponding reasoning confidence is high. Since the reasoning VLM lacks direct confidence, we asked it to output a confidence score (0–100) based on its own reasoning. We apply a hard threshold $\sigma$: if the local VLM's confidence is $\geq \sigma$, we use the expert model's answer; otherwise, we use the coordinator's final answer. We report this final performance alongside our CARE-Flow-B in the Tab. 15. We observe that adapting the final answer based directly on the expert VLM's confidence score generally does not improve performance, consistent with previous evaluations. Our coordinator's CoT review process already implicitly considers the confidence of the expert VLM's reasoning trace, as we ask it

Table 13: **Full Benchmarking Results on Medical VQA Datasets.** We present the full benchmarking results on 4 Medical VQA datasets (Hu et al., 2024; Liu et al., 2021; Ben Abacha et al., 2019; Lau et al., 2018) and report their accuracy (%). The open-ended questions are evaluated using GPT-4o (Hurst et al., 2024) against ground-truth. We use the Instruct fine-tuned model whenever available. We use a $3k$ subset of OmniMedVQA (OMVQA) Hu et al. (2024) dataset for benchmark. We highlight results of medical expert VLMs using gray, and our model using green.

| Model | OMVQA-3k | VQA-RAD | SLAKE | VQA-Med-2019 | Overall |
|---|---|---|---|---|---|
| *Proprietary* | | | | | |
| GPT-4o (Hurst et al., 2024) | 64.07 | 58.54 | 63.55 | 59.60 | 61.44 |
| GPT-5-mini (OpenAI, 2025) | 68.57 | 59.87 | 65.94 | 61.60 | 63.99 |
| GPT-5 (OpenAI, 2025) | 74.73 | 63.19 | 67.75 | 62.20 | 66.97 |
| *Open-source* | | | | | |
| Llama-3.2-11B-Vision (Meta, 2024) | 43.10 | 53.22 | 63.17 | 57.40 | 54.22 |
| Qwen2.5-VL-3B (Bai et al., 2025) | 57.07 | 54.77 | 53.24 | 43.00 | 52.02 |
| Qwen2.5-VL-7B (Bai et al., 2025) | 61.40 | 54.10 | 59.73 | 50.60 | 56.46 |
| Qwen2.5-VL-32B (Bai et al., 2025) | 65.10 | 61.20 | 65.46 | 51.60 | 60.84 |
| Qwen2.5-VL-72B (Bai et al., 2025) | 65.27 | 62.75 | 67.37 | 53.00 | 62.10 |
| InternVL3-2B (Zhu et al., 2025a) | 75.43 | 55.65 | 63.07 | 54.00 | 62.04 |
| InternVL3-8B (Zhu et al., 2025a) | 75.97 | 61.86 | 66.13 | 57.40 | 65.34 |
| InternVL3-14B (Zhu et al., 2025a) | 77.70 | 63.64 | 68.61 | 57.60 | 66.89 |
| InternVL3-38B (Zhu et al., 2025a) | 78.57 | 62.97 | 68.70 | 58.80 | 67.26 |
| InternVL3-78B (Zhu et al., 2025a) | 80.73 | 65.85 | 72.42 | **61.40** | 70.10 |
| DeepEyes-7B (Zheng et al., 2025) | 57.40 | 56.10 | 61.16 | 52.20 | 56.72 |
| llava-med-v1.5-mistral-7b (Li et al., 2023) | 45.30 | 41.91 | 50.86 | 37.00 | 43.77 |
| MedVLM-R1-2B (Pan et al., 2025) | 72.07 | 41.46 | 46.47 | 45.40 | 51.35 |
| medgemma-4b (Sellergren et al., 2025) | 61.50 | 58.09 | 69.66 | 47.40 | 59.16 |
| medgemma-27b (Sellergren et al., 2025) | 64.23 | 62.75 | 70.52 | 48.40 | 61.47 |
| HuatuoGPT-Vision-7B (Chen et al., 2024a) | 70.70 | 59.87 | 60.50 | 57.20 | 62.07 |
| HuatuoGPT-Vision-34B (Chen et al., 2024a) | 76.80 | 60.75 | 64.12 | 60.60 | 65.57 |
| Lingshu-7B (Xu et al., 2025) | 73.17 | 58.54 | 76.15 | 58.80 | 66.66 |
| Lingshu-32B (Xu et al., 2025) | 83.97 | 64.75 | 82.25 | 58.20 | 72.29 |
| **CARE-Flow-S (4B)** | 94.53 | 56.32 | 78.44 | 53.60 | 70.72 |
| **CARE-Coord-S** | 97.70 | 62.75 | 77.19 | 60.60 | 74.56 |
| **CARE-Flow-B (10B)** | 96.17 | 63.64 | **83.21** | 56.60 | 74.91 |
| **CARE-Coord-B** | **97.97** | **68.29** | 83.11 | 60.80 | **77.54** |

Table 14: **Comparison with Fine-tuned Baseline.** We report medical VQA accuracy (%) on four standard benchmarks: three in-domain OMVQA-3k (Hu et al., 2024), VQA-RAD (Lau et al., 2018), SLAKE (Liu et al., 2021) and one out-of-domain VQA-Med-2019 (Ben Abacha et al., 2019). We compare with the InternVL3 (Zhu et al., 2025a) baseline fine-tuned with the same training data to isolate the influence of different training data. Our results are highlighted in blue.

| Model | OMVQA-3k | VQA-RAD | SLAKE | VQA-Med-2019 | Overall |
|---|---|---|---|---|---|
| InternVL3-2B (zero-shot) | 75.43 | 55.65 | 63.07 | 54.00 | 62.04 |
| InternVL3-2B-Finetuned | 87.97 | 57.43 | 69.56 | 51.00 | 66.49 |
| **CARE-Flow-S** | 94.53 | 56.32 | 78.44 | 53.60 | 70.72 |
| InternVL3-8B (zero-shot) | 75.97 | 61.86 | 66.13 | **57.40** | 65.34 |
| InternVL3-8B-Finetuned | 91.13 | 61.86 | 76.53 | 53.80 | 70.83 |
| **CARE-Flow-B** | **96.17** | **63.64** | **83.21** | 56.60 | **74.91** |

to review the CoT quality. Consequently, the coordinator insists on the expert VLM's output when its reasoning trace is confident.

**LLM-as-Judge Stability.** We also evaluate our results with different LLM-as-judges (both proprietary (Hurst et al., 2024) and open-source (Zhu et al., 2025a)) to demonstrate the variance stemming from the judger. We skip OmniMedVQA (Hu et al., 2024) in this evaluation as it only contains closed-ended questions. We report the performance in Tab. 16. We note that the variance between different LLM-as-judges is very small (less than 1%). Our performance improvement is significant enough considering this variance.

Table 15: **Conservative Coordinator Review Strategy.** We report medical VQA accuracy (%) on four standard benchmarks (Hu et al., 2024; Lau et al., 2018; Liu et al., 2021; Ben Abacha et al., 2019) to compare our coordinator strategy against a conservative strategy that votes for EG-VLM's answer when its confidence is high. We report the performance with different confidence threshold $\sigma$. Our results are highlighted in green.

| Model | OMVQA-3k | VQA-RAD | SLAKE | VQA-Med-2019 | Overall |
|---|---|---|---|---|---|
| $\sigma = 25$ | 96.70 | 66.29 | 83.77 | 56.40 | 75.79 |
| $\sigma = 50$ | 96.74 | 67.40 | **83.87** | 56.60 | 76.15 |
| $\sigma = 75$ | 97.47 | 67.85 | 83.68 | 56.80 | 76.45 |
| **CARE-Coord-B** | **97.97** | **68.29** | 83.11 | **60.80** | **77.54** |

Table 16: **Different LLM-as-Judge.** We report the performance on the datasets with open-ended questions Lau et al. (2018); Liu et al. (2021); Ben Abacha et al. (2019) to demonstrate the variance of different LLM-as-judge. We experimented with both proprietary (Hurst et al., 2024) and open-source (Zhu et al., 2025a) LLMs of different sizes. We report the averaged performance across different LLMs and the corresponding standard deviation.

| LLM Judger | VQA-RAD | SLAKE | VQA-Med-2019 | Overall |
|---|---|---|---|---|
| GPT-4o | 68.29 | 83.11 | 60.80 | 70.73 |
| GPT-4o-mini | 68.51 | 84.35 | 60.60 | 71.15 |
| InternVL3-38B-Instruct | 67.41 | 82.44 | 59.00 | 69.62 |
| InternVL3-78B-Instruct | 67.41 | 83.30 | 60.20 | 70.30 |
| Avg. | 67.91 | 83.30 | 60.15 | 70.45 |
| STD | $\pm0.58$ | $\pm0.79$ | $\pm0.81$ | $\pm0.66$ |

**Fine-tuning Strategy for Entity Proposal Model.** We report the performance of the entity proposal VLM with SFT and SFT + RFT to validate our choice of using RFT on the entity proposal VLM. We use the static CARE-Flowframework here to isolate the influence of the coordinator review. As shown Tab. 17, using SFT alone or combining SFT and RFT underperforms our RFT model, which validates our choice of training recipe. This suggests that applying SFT with synthetic data may be harmful to our task. Applying RFT over SFT indeed helps general performance, but for a task that is already within the model's capability, directly applying RFT could be the best option.

### D.7 HUMAN EVALUATION

To better quantify the accountability of our method, we conduct a human evaluation to assess the quality of our model's reasoning trace.

**Experimental Setting.** We randomly sample 35 correctly answered cases from the four test datasets to evaluate the accountability of the reasoning traces. We develop a web-based evaluation platform, where human evaluators can examine the full reasoning process and assign a True/False judgment for each case (the user study interface is shown in Fig. 24). We recruit nine medical students to perform the evaluations through the platform. For comparison, we also included GPT-4o as the coordinator baseline.

**Experiment Participant.** Due to the limited time, we are only able to contact participants with PhD/MD-level knowledge for our experiments. We gathered 9 pieces of feedback from 9 participants with no prior knowledge about our work, all of whom have either obtained or are pursuing a PhD/MD degree related to the medical and imaging domain. We plan to further collaborate with experts with clinical experience in future research, and we agree that this is a critical step towards real-world application.

**Results.** We report the human evaluation pass rate alongside their original overall medical VQA performance in Tab. 18. CARE-Coord-B achieved a human evaluation pass rate of 82.14%, surpassing the GPT-4o baseline (73.94%). This result demonstrates that our proposed framework not only achieves higher accuracy but, more importantly, generates reasoning traces that are more factually accurate and visually grounded, thereby offering superior clinical accountability.

Table 17: **Entity Proposal Model Fine-tuning Strategy Ablation.** We ablate different fine-tuning strategies for the Entity proposal VLM on the entity proposal, referring entity segmentation, and medical VQA tasks. Our choice of training strategy is highlighted in blue.

| Method | Entity Accuracy | Mask Dice | Overall MedVQA |
|---|---|---|---|
| SFT | 76.70 | 72.69 | 74.26 |
| SFT + RFT | 80.92 | **74.03** | 74.44 |
| RFT (CARE-Flow-B) | **85.28** | 73.48 | **74.91** |

Table 18: **Reasoning Trace Human Evaluation.** We conduct a human evaluation that evaluates the quality of the reasoning trace of our model on a subset of samples. We report the pass rate of human evaluation. We compare our CARE-Coord-B against our model with the GPT-4o coordinator. Our method is highlighted in green.

| Coordinator | Human Evaluation Pass Rate (%) |
|---|---|
| **GPT-4o** | 73.94 |
| **GPT-5 (CARE-Coord-B)** | 82.14 |

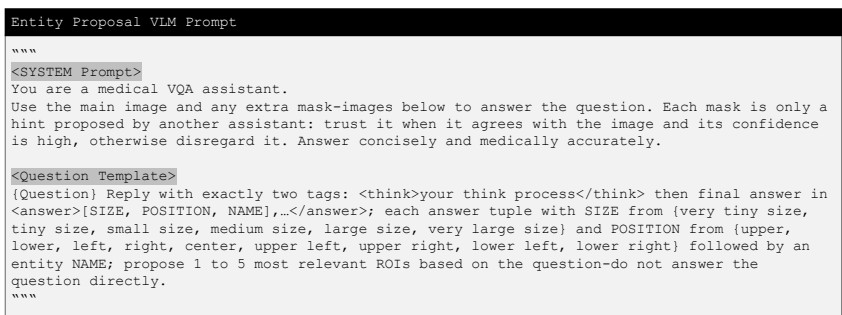

Figure 8: **System Prompt for Entity Proposal Model.** We present the system prompt for our entity proposal model, where we instruct the model to name the most relevant medical entity related to the user question. We note that even if we asked the model to also generate the size and position of the entity, we only use the entity name for the downstream tasks.

## D.8 ENTITY PROPOSAL MODEL PROMPT

We provide the prompt for the entity proposal model in Fig. 8. Notably, we ask the model to generate the size and position information of the proposed entity, but we choose not to use it during the segmentation and downstream inference, as we notice this information can introduce unexpected hallucination, as the spatial reasoning capability is not a strength of our base model. Still, including this information can serve as a self-prompting and help generate the final entity proposal. Meanwhile, the size and position information also do not participate in the calculation of the similarity reward; we only use the entity name in Eq. (1).

## D.9 EG-VQA MODEL PROMPT

We present the prompt of the EG-VQA model in Fig. 9, where we mainly highlight the meaning of each type of visual clue. This can help the model better understand the properties of different visual clues and focus on the corresponding region or local detail when fed with a different clue.

## D.10 COORDINATOR PROMPT

We provide the full prompt used for the coordinator in Fig. 10. We define the behavior of the coordinator model in the system prompt and instruct it to plan and review. We instruct the coordinator model to review and double-check if the chain of thought and the answer from the tool VLM are consistent, and ask it to correct the answer when necessary. We also require the coordinator to always

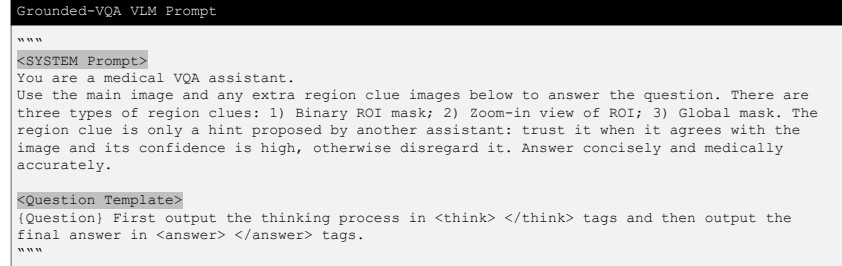

Figure 9: **System Prompt for G-VQA Model.** We present the system prompt for our grounded VQA model, where we provide information about different types of visual clues and guide the model to focus differently when given these visual clues.

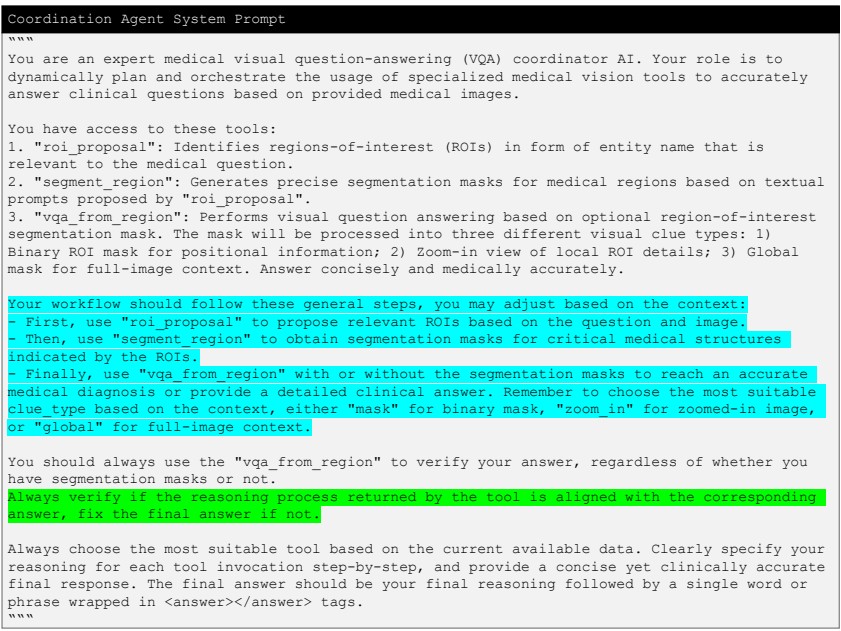

Figure 10: **System Prompt for Coordinator Model.** We present the system prompt for our coordinator model. We introduce the overview of each tool and define the general workflow here. The section highlighted in blue defines the coordination and planning behavior, and the section highlighted in green defines the iterative answer review process during inference.

make at least one tool call to the EG-VQA model, as the EG-VQA model is more convincing on the medical task, and the coordinator model is only for action planning and answer review.

We also provide the tool schema to the coordinator model during inference, which is defined in Appendix D.13.

### D.11 EXAMPLE SYNTHETIC ROI PROPOSAL DATA

We provide randomly sampled synthetic data examples in Figs. 11 and 12. The full data will be released upon acceptance.

### D.12 MORE CASE STUDY

We provide randomly sampled cases during model inference using CARE-Coord-B in Figs. 14 to 19. We also include an example of common failure cases in Fig. 20, where the reason for failure is due to hallucination introduced by the coordinator. And the coordinator over-edited the answer from the model to insist on the wrong answer.

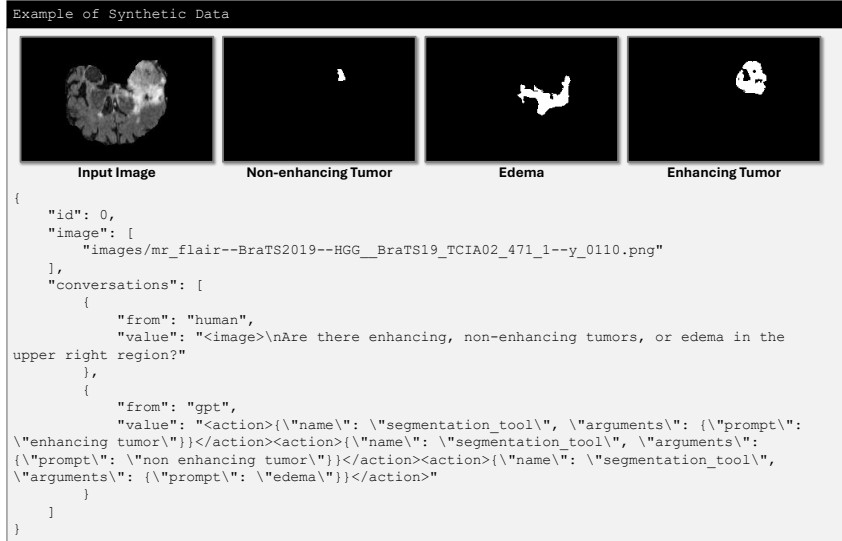

Figure 11: **Example of Synthetic Data for Entity Proposal Task.** An example of synthetic data for the entity proposal task, we wrap the ground truth entity name in the JSON object.

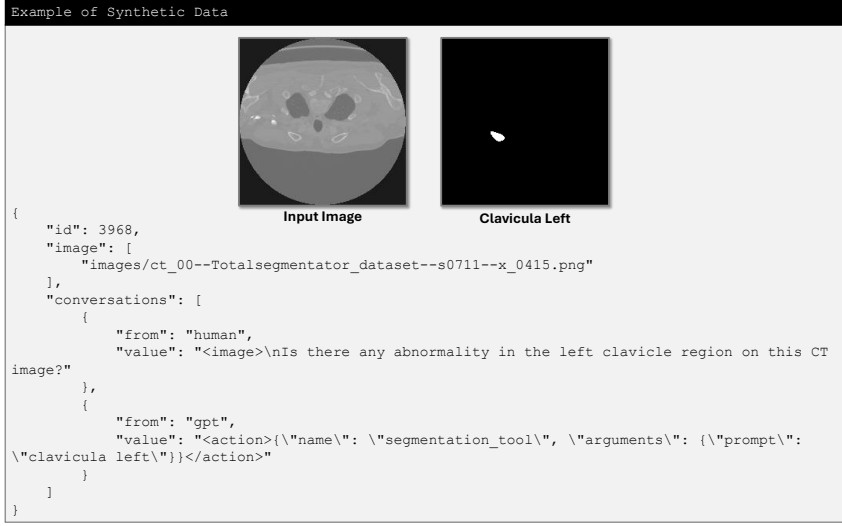

Figure 12: **Example of Synthetic Data for Entity Proposal Task.** An example of synthetic data for the entity proposal task, we wrap the ground truth entity name in the JSON object.

### D.13 TOOL FUNCTION DEFINITION

We provide the function definition for the coordinator model in Figs. 21 to 23.

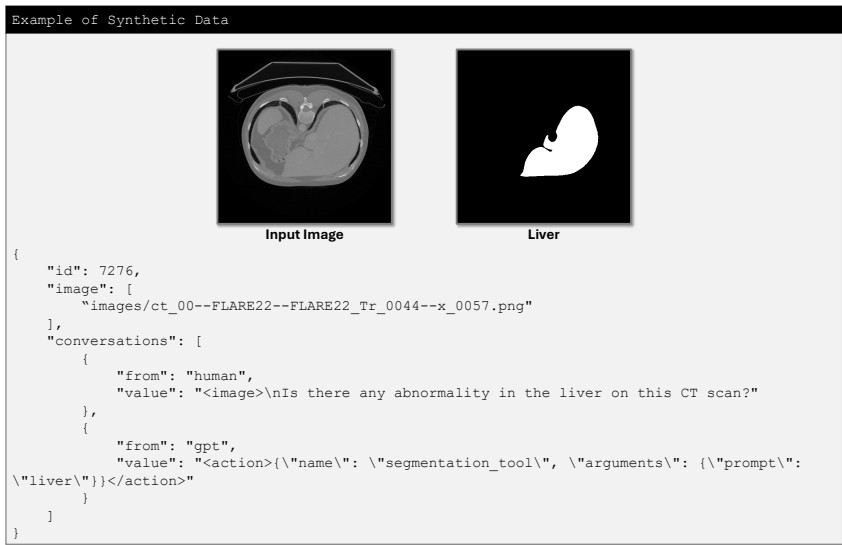

Figure 13: **Example of Synthetic Data for Entity Proposal Task.** An example of synthetic data for the entity proposal task, we wrap the ground truth entity name in the JSON object.

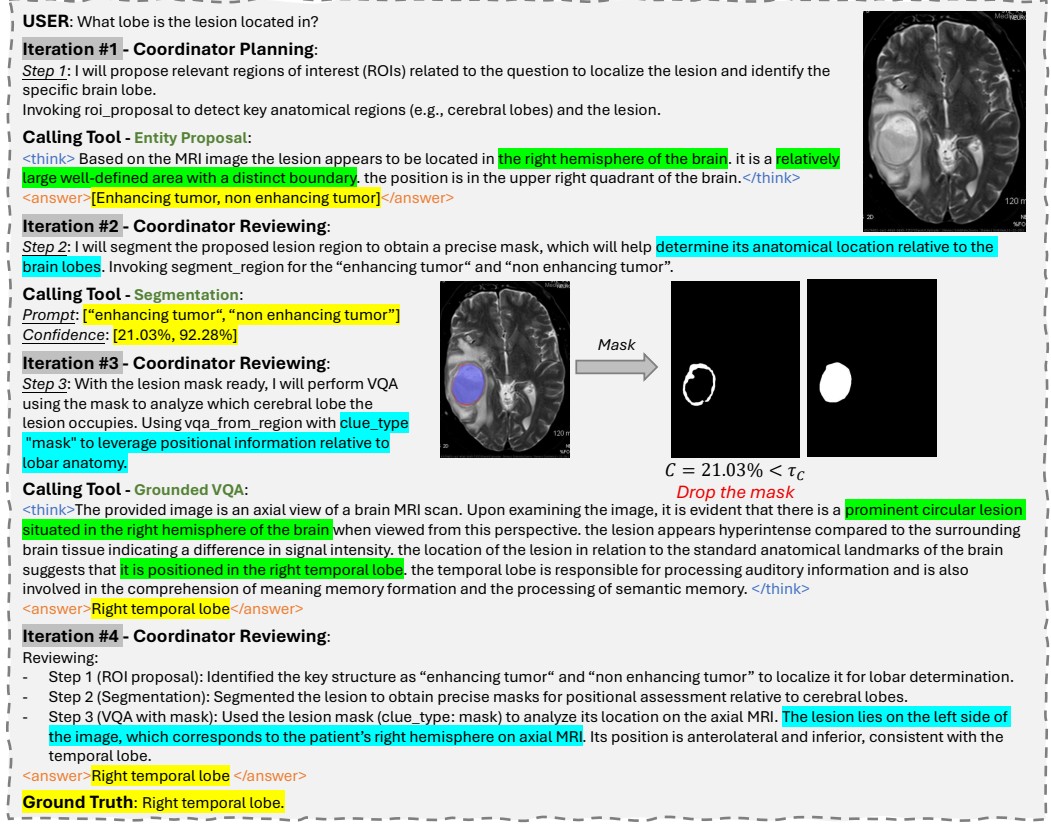

Figure 14: **More Case Study.** We provide a full reasoning process of a user's question about a brain MRI image here. We highlight the key information from the coordinator in blue and key information from the tool in green.

**USER**: Is there airspace consolidation on the left side?

**Iteration #1 - Coordinator Planning**:

*Step 1*: I will propose relevant regions of interest (ROIs) based on the question to focus on the left lung fields where airspace consolidation would appear.
Proceeding to ROI proposal.

**Calling Tool - Entity Proposal**:

<think> Based on the x-ray image I will analyze the left side for any signs of airspace consolidation. I will look for areas of increased opacity that may indicate consolidation. I will also consider the position and size of these areas.</think>

<answer>[Airspace Consolidation]</answer>

**Iteration # 2 - Coordinator Reviewing**:

*Step 2*: The initial ROI term was too generic. To accurately assess airspace consolidation on the left side, I will segment the anatomical left lung to focus analysis on the relevant parenchyma. Proceeding to segment the left lung.

**Calling Tool - Segmentation**:
*Prompt*: ["left side lung"]
*Confidence*: [94.73%]

**Iteration #3 - Coordinator Reviewing**:

*Step 3*: I will perform VQA focused on the segmented left lung using a zoom-in view to best evaluate parenchymal details for airspace consolidation.

**Calling Tool - Grounded VQA**:

<think>The provided chest X-ray image displays an anteroposterior view of the thoracic cavity highlighting the lungs heart and surrounding structures. Upon examination the left lung field appears to have an area of increased opacity compared to the right lung suggesting the presence of airspace consolidation. This is indicated by the denser whiter region within the left lung which contrasts with the typically darker air-filled lung fields. The heart and mediastinal structures are centrally located and the diaphragm appears intact. The increased opacity on the left side is consistent with fluid cells or other material filling the alveolar spaces a characteristic feature of airspace consolidation.</think>

<answer>Yes</answer>

**Iteration #4 - Coordinator Reviewing**:

Reviewing :
- ROI proposal: Targeted the region relevant to the question parenchyma where airspace consolidation would manifest.
- Segmentation: Isolated the left lung to restrict analysis to the correct side (mask confidence ~95%).
- VQA with zoom-in: Focused on parenchymal details within the left lung; the zoomed assessment showed a region of increased opacity consistent with alveolar filling, supporting airspace consolidation.

<answer>Yes</answer>

**Ground Truth**: Yes

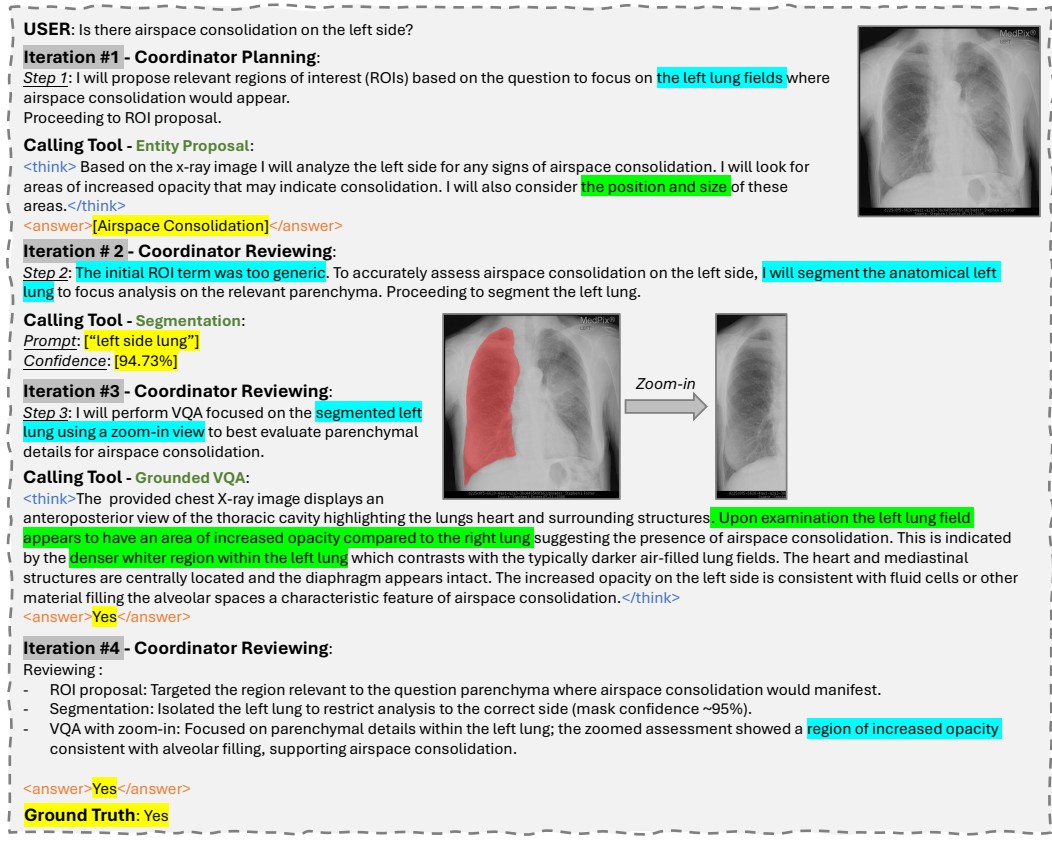

Figure 15: **More Case Study.** We provide a full reasoning process of a user's question about a Chest X-ray image here. We highlight the key information from the coordinator in blue and key information from the tool in green.

**USER**: Is there any anomaly or deviation detectable in this picture?
   A: No       B: Yes.

**Iteration #1 - Coordinator Planning**:

*Step 1*: Start with global clue to assess overall normalcy, then confirm with segmentation mask if needed. Proceeding with VQA using global context first.

**Calling Tool - Grounded VQA**:

<think>The provided image is a transverse section from a CT scan of the abdomen. It clearly shows various anatomical structures including the vertebra in the center surrounded by soft tissues and organs such as the intestines. The image appears to be a normal abdominal scan with no obvious signs of abnormalities like tumors significant fluid accumulation or unusual masses. The vertebra and surrounding bones appear intact and the soft tissues do not show any apparent signs of distress or irregularities. The distribution and density of the tissues seem consistent with typical abdominal anatomy.</think>

<answer>A</answer>

**Iteration #2 - Coordinator Reviewing**:

Reviewing :
- VQA verification: Used global context first to evaluate the entire abdomen for any anomalies, as the question is about any deviation. The VQA assessment of the global image indicated no abnormality.
- Conclusion: No detectable anomaly on this abdominal CT slice.

<answer>A. No</answer>

**Ground Truth**: A.

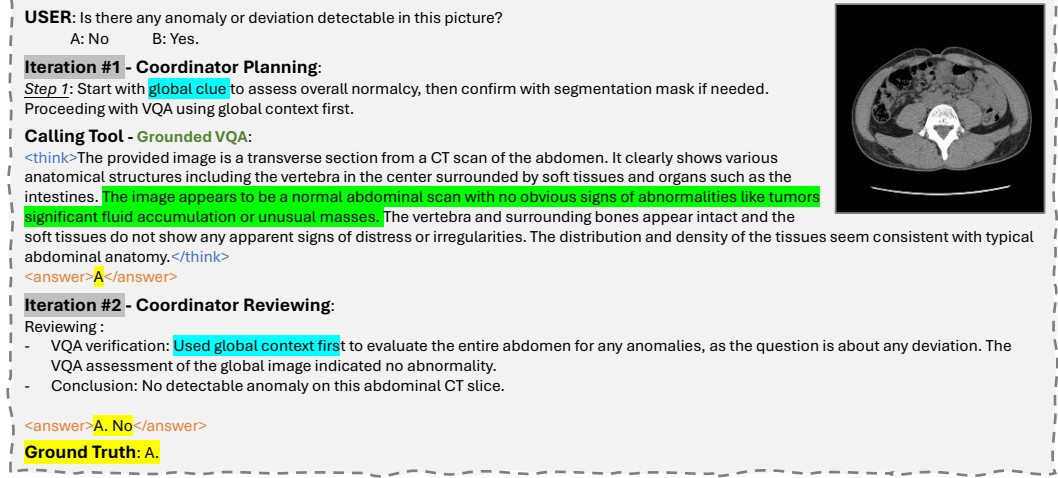

Figure 16: **More Case Study.** We provide a full reasoning process of a user's question about a body CT image here. We highlight the key information from the coordinator in blue and key information from the tool in green. This example skipped the entity proposal and segmentation, as the question is about global information of the image.

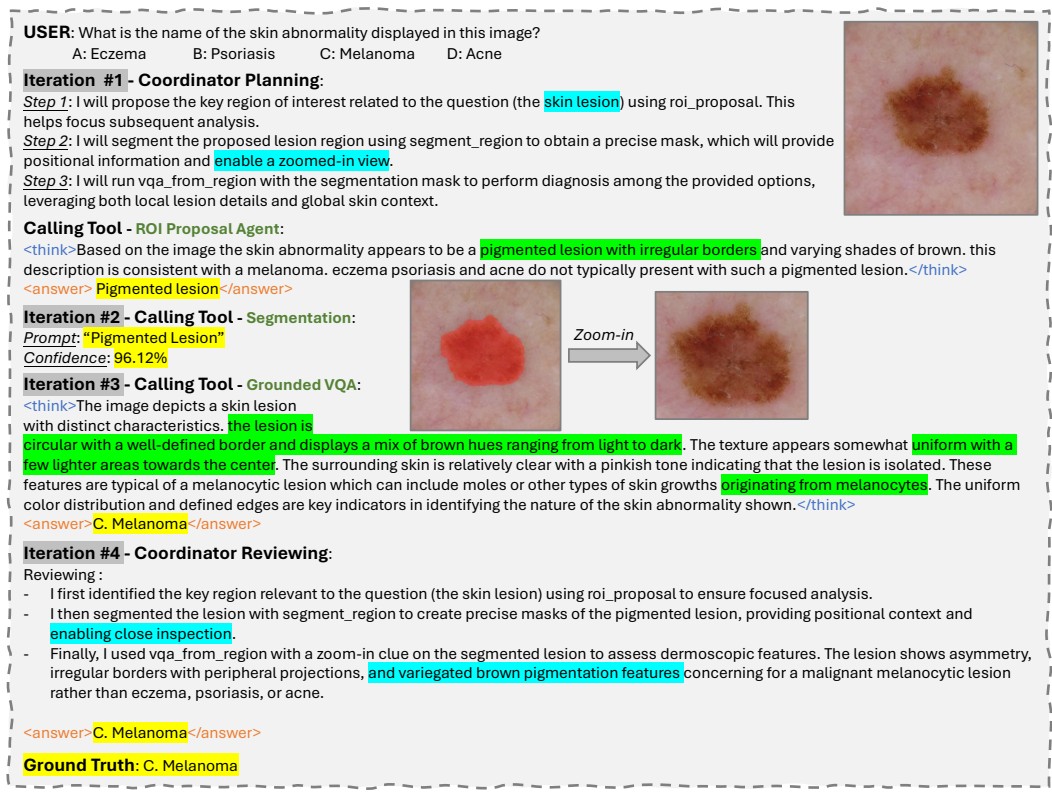

**USER**: What is the name of the skin abnormality displayed in this image?

       A: Eczema      B: Psoriasis      C: Melanoma      D: Acne

**Iteration #1 - Coordinator Planning**:

*Step 1*: I will propose the key region of interest related to the question (the skin lesion) using roi_proposal. This helps focus subsequent analysis.

*Step 2*: I will segment the proposed lesion region using segment_region to obtain a precise mask, which will provide positional information and enable a zoomed-in view.

*Step 3*: I will run vqa_from_region with the segmentation mask to perform diagnosis among the provided options, leveraging both local lesion details and global skin context.

**Calling Tool - ROI Proposal Agent**:

<think>Based on the image the skin abnormality appears to be a pigmented lesion with irregular borders and varying shades of brown. this description is consistent with a melanoma. eczema psoriasis and acne do not typically present with such a pigmented lesion.</think>

<answer>Pigmented lesion</answer>

**Iteration #2 - Calling Tool - Segmentation**:

*Prompt*: "Pigmented Lesion"

*Confidence*: 96.12%

**Iteration #3 - Calling Tool - Grounded VQA**:

<think>The image depicts a skin lesion with distinct characteristics. the lesion is circular with a well-defined border and displays a mix of brown hues ranging from light to dark. The texture appears somewhat uniform with a few lighter areas towards the center. The surrounding skin is relatively clear with a pinkish tone indicating that the lesion is isolated. These features are typical of a melanocytic lesion which can include moles or other types of skin growths originating from melanocytes. The uniform color distribution and defined edges are key indicators in identifying the nature of the skin abnormality shown.</think>

<answer>C. Melanoma</answer>

**Iteration #4 - Coordinator Reviewing**:

Reviewing :

- I first identified the key region relevant to the question (the skin lesion) using roi_proposal to ensure focused analysis.
- I then segmented the lesion with segment_region to create precise masks of the pigmented lesion, providing positional context and enabling close inspection.
- Finally, I used vqa_from_region with a zoom-in clue on the segmented lesion to assess dermoscopic features. The lesion shows asymmetry, irregular borders with peripheral projections, and variegated brown pigmentation features concerning for a malignant melanocytic lesion rather than eczema, psoriasis, or acne.

<answer>C. Melanoma</answer>

**Ground Truth**: C. Melanoma

Figure 17: **More Case Study.** We provide a full reasoning process of a user's question about a digital image here. We highlight the key information from the coordinator in blue and key information from the tool in green.

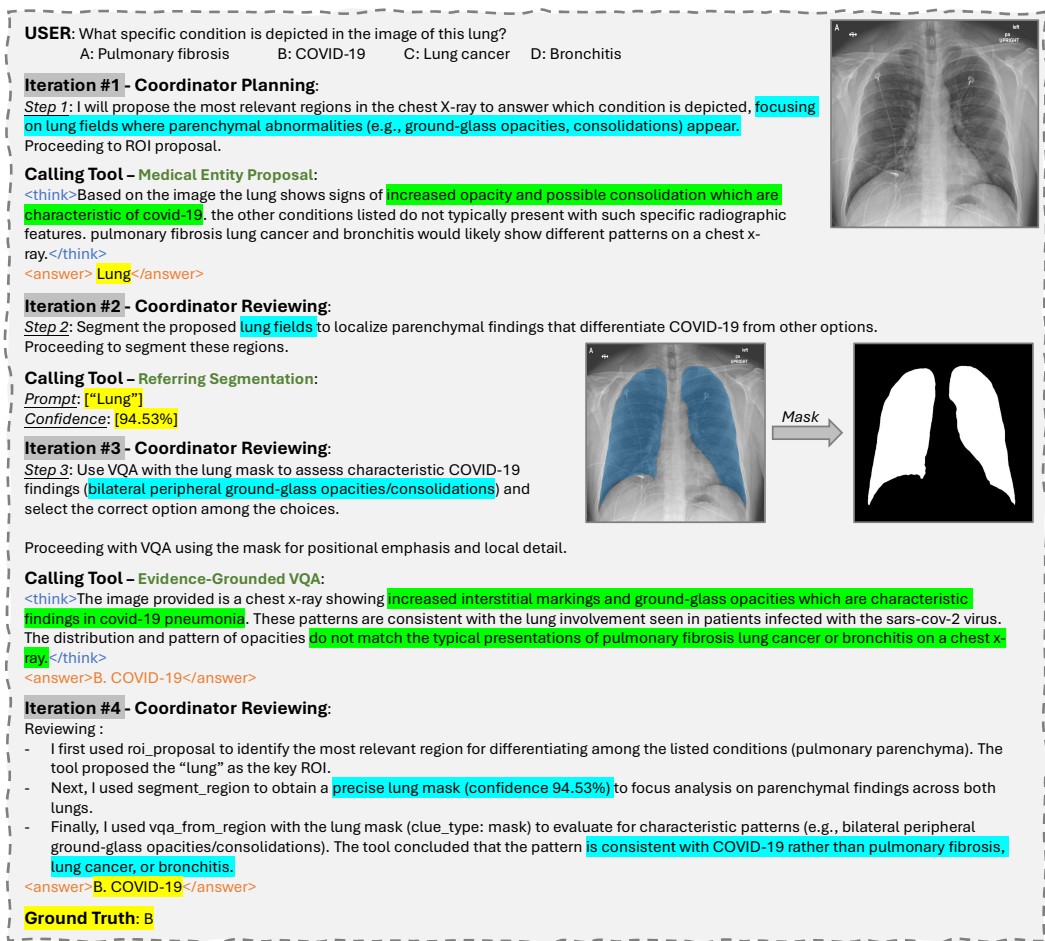

**USER**: What specific condition is depicted in the image of this lung?
A: Pulmonary fibrosis    B: COVID-19    C: Lung cancer    D: Bronchitis

**Iteration #1 - Coordinator Planning**:
*Step 1*: I will propose the most relevant regions in the chest X-ray to answer which condition is depicted, focusing on lung fields where parenchymal abnormalities (e.g., ground-glass opacities, consolidations) appear.
Proceeding to ROI proposal.

**Calling Tool – Medical Entity Proposal**:
<think>Based on the image the lung shows signs of increased opacity and possible consolidation which are characteristic of covid-19. the other conditions listed do not typically present with such specific radiographic features. pulmonary fibrosis lung cancer and bronchitis would likely show different patterns on a chest x-ray.</think>
<answer> Lung</answer>

**Iteration #2 - Coordinator Reviewing**:
*Step 2*: Segment the proposed lung fields to localize parenchymal findings that differentiate COVID-19 from other options.
Proceeding to segment these regions.

**Calling Tool – Referring Segmentation**:
*Prompt*: ["Lung"]
*Confidence*: [94.53%]

**Iteration #3 - Coordinator Reviewing**:
*Step 3*: Use VQA with the lung mask to assess characteristic COVID-19 findings (bilateral peripheral ground-glass opacities/consolidations) and select the correct option among the choices.

Proceeding with VQA using the mask for positional emphasis and local detail.

**Calling Tool – Evidence-Grounded VQA**:
<think>The image provided is a chest x-ray showing increased interstitial markings and ground-glass opacities which are characteristic findings in covid-19 pneumonia. These patterns are consistent with the lung involvement seen in patients infected with the sars-cov-2 virus. The distribution and pattern of opacities do not match the typical presentations of pulmonary fibrosis lung cancer or bronchitis on a chest x-ray.</think>
<answer>B. COVID-19</answer>

**Iteration #4 - Coordinator Reviewing**:
Reviewing :
- I first used roi_proposal to identify the most relevant region for differentiating among the listed conditions (pulmonary parenchyma). The tool proposed the "lung" as the key ROI.
- Next, I used segment_region to obtain a precise lung mask (confidence 94.53%) to focus analysis on parenchymal findings across both lungs.
- Finally, I used vqa_from_region with the lung mask (clue_type: mask) to evaluate for characteristic patterns (e.g., bilateral peripheral ground-glass opacities/consolidations). The tool concluded that the pattern is consistent with COVID-19 rather than pulmonary fibrosis, lung cancer, or bronchitis.
<answer>B. COVID-19</answer>

**Ground Truth**: B

Figure 18: **More Case Study.** We provide a full reasoning process of a user's question about a Chest X-ray here. We highlight the key information from the coordinator in blue and key information from the tool in green.

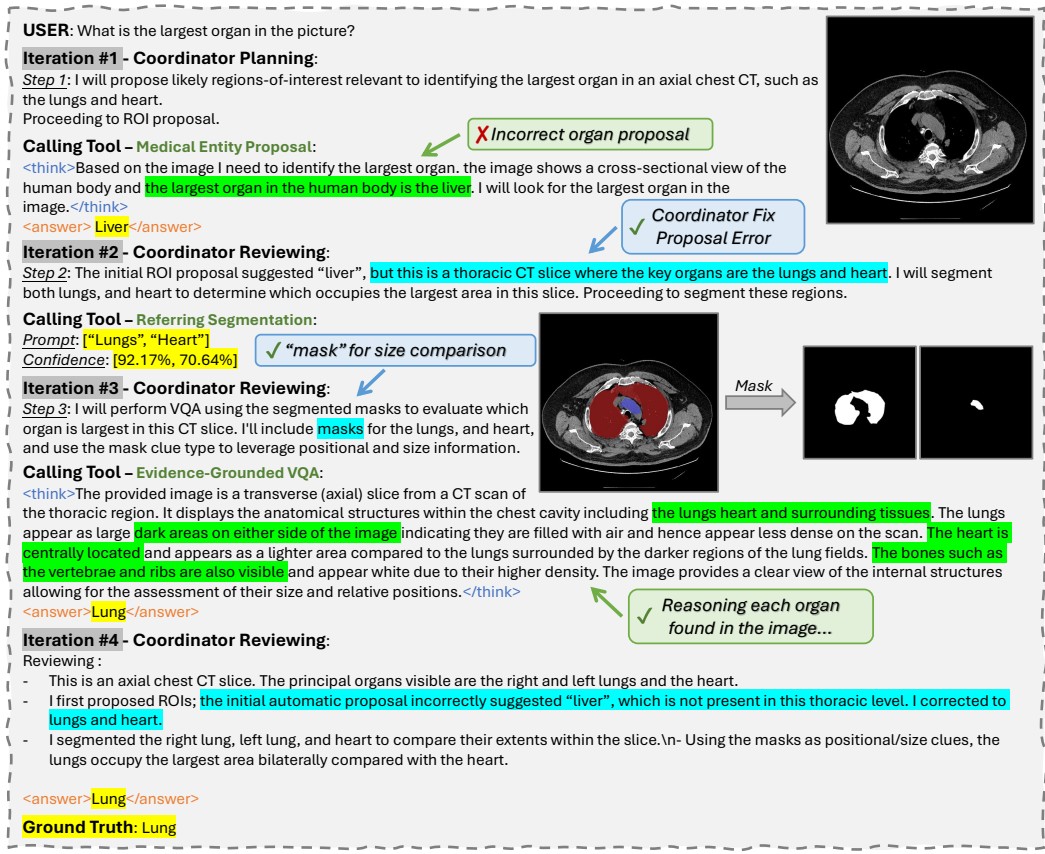

Figure 19: **Case Study.** We present the complete reasoning trace for a CT question. Key information from the coordinator is highlighted in blue, model reasoning in green, and each model's final answer in yellow. Notably, during review, the coordinator corrects an initial error from the entity proposal.

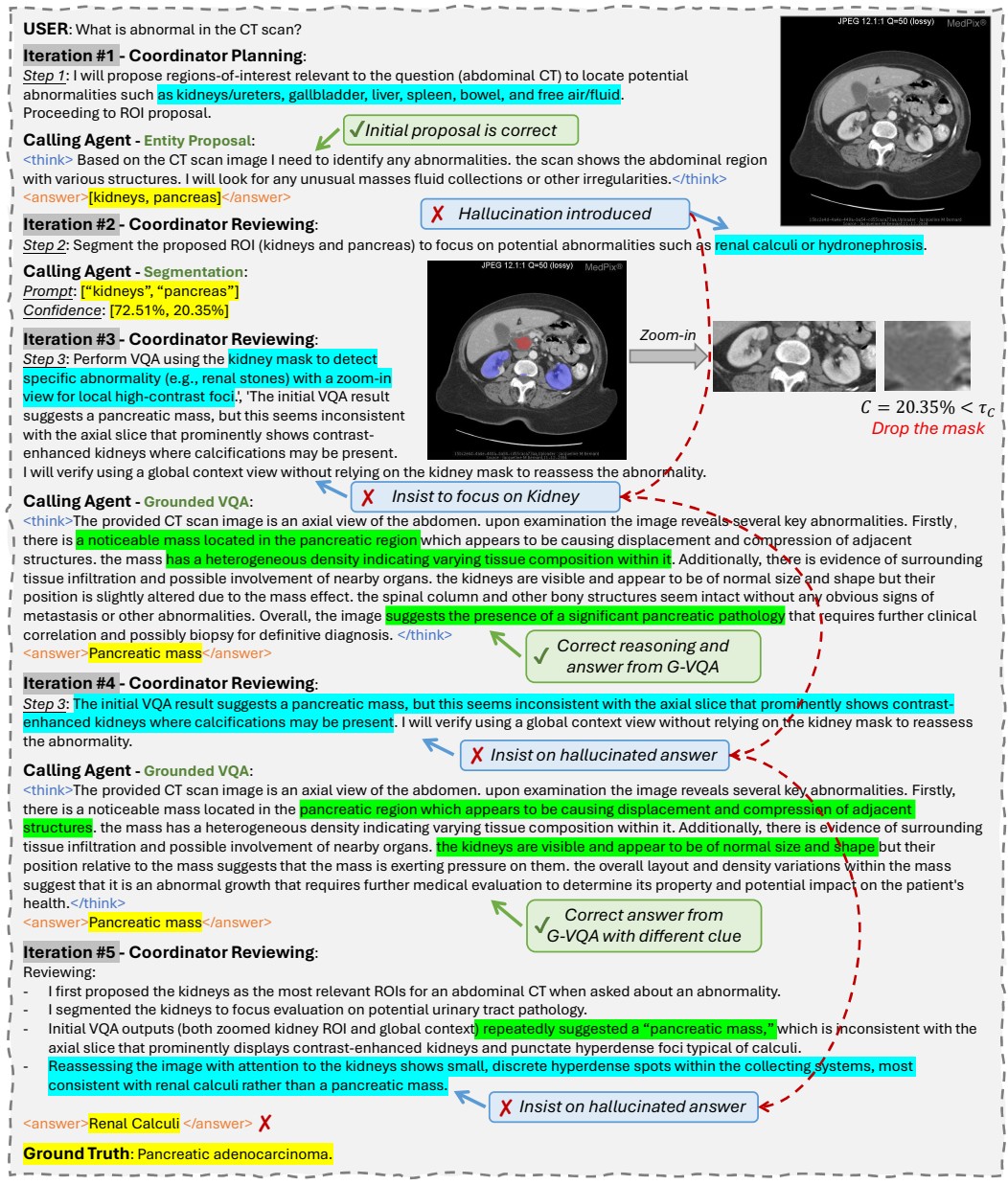

Figure 20: **Failed Case Study.** We provide a full reasoning process of a user's question about a CT image with a failed result here. We highlight the reason for failure during the reasoning chain, where the G-VQA model gives a correct answer, but the coordinator model insists on a hallucinated answer. We use the red dashed arrows to illustrate how the hallucination propagates during the inference.

```
Entity Proposal Function Schema

{
    "type": "function",
    "function": {
        "name": "roi_proposal",
        "description": "Proposes medically relevant regions (entity name) in the medical image
based on the question.",
        "parameters": {
            "type": "object",
            "properties": {
                "image_url": {"type": "string"},
                "question": {"type": "string"}
            },
            "required": ["image_url", "question"],
        }
    }
}
```

Figure 21: **Function Definition for Entity Proposal Model** We provide the full definition of the function schema of the entity proposal model used during coordination. We explain in detail the definition of each parameter here.

```
Segmentation Function Schema

{
    "type": "function",
    "function": {
        "name": "segment_region",
        "description": "Segments specific medical regions from medical image given textual
prompt.",
        "parameters": {
            "type": "object",
            "properties": {
                "image_url": {"type": "string"},
                "prompt": {
                    "type": "array",
                    "items": {
                        "type": "string"
                    },
                    "description": "List of medical entity (e.g., lung, liver)"
                }
            },
            "required": ["image_url", "prompt"],
        }
    }
}
```

Figure 22: **Function Definition for Segmentation Model** We provide the full definition of the function schema of the referring segmentation model, which accepts two inputs and outputs the segmentation mask along with its confidence.

```
G-VQA Function Schema
{
    "type": "function",
    "function": {
        "name": "vqa_from_region",
        "description": "Performs medical visual question answering based on optional visual
clue image (either global view indicator, binary mask or zoom-in ROI).",
        "parameters": {
            "type": "object",
            "properties": {
                "image_url": {"type": "string"},
                "question": {"type": "string"},
                "mask_url": {
                    "type": "array",
                    "items": {
                        "type": "string"
                    },
                    "description": "List of binary mask path of segmented region"
                },
                "mask_name": {
                    "type": "array",
                    "items": {
                        "type": "string"
                    },
                    "description": "List of entity name of the mask, e.g., 'lung', 'liver',
'tumor'"
                },
                "mask_confidence": {
                    "type": "array",
                    "items": {
                        "type": "number",
                        "minimum": 0,
                        "maximum": 100
                    },
                    "description": "List of confidence score of each mask, between 0 and 100%"
                },
                "clue_type": {
                    "type": "string",
                    "enum": ["global", "mask", "zoom_in"],
                    "description": "Type of visual clue to provide: 'global' for global
context indicator, 'mask' for binary mask, 'zoom_in' for zoomed-in image. Global view
highlights overall context, binary mask highlights positional information and zoom-in view
highlights local details."
                }
            },
            "required": ["image_url", "question", "clue_type"],
        }
    }
}
```

Figure 23: **Function Definition for G-VQA Model** We provide the full definition of the function schema of the G-VQA model. Note that we only set the input image, question, and clue type to be required parameters, as when using a global clue type, there is no need to use other parameters about the mask.

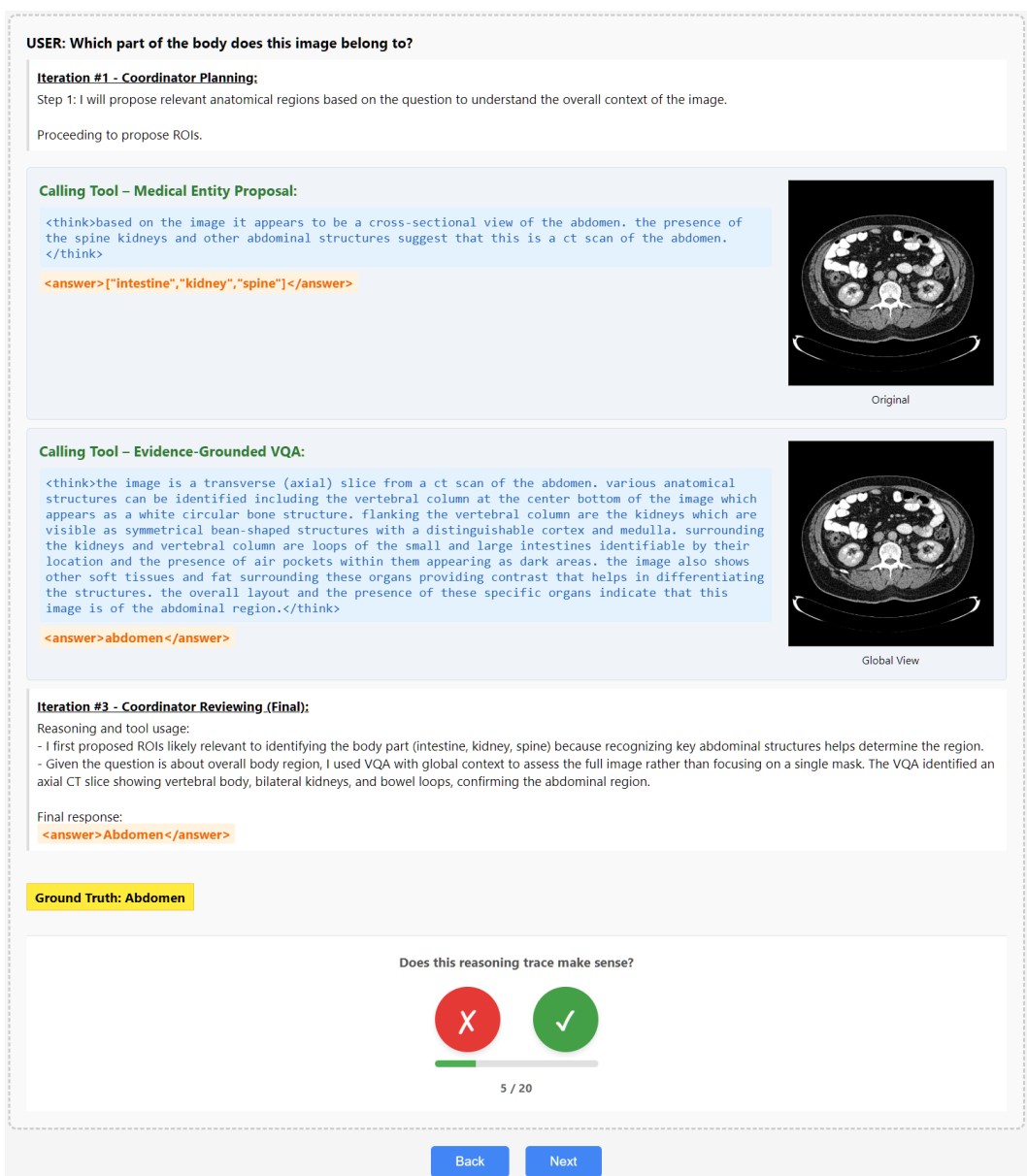

Figure 24: **User Interface of Human Evaluation.** We provide the full user interface of our human evaluation.

