# OpenReview forum: "CARE: Towards Clinical Accountability in Multi-Modal Medical Reasoning with an Evidence-Grounded Agentic Framework"
_ICLR.cc/2026/Conference — ICLR 2026 Poster_

### Official Review · Reviewer_fS8R · 2025-10-20

**Soundness:** 3
**Presentation:** 4
**Contribution:** 3
**Rating:** 6
**Confidence:** 4

**Summary:**

The paper proposes a framework for multi-modal reasoning specifically designed for medical VQA tasks. The task is split into three separate stages: (i) identifying medical entities relevant to the question, (ii) segmenting those entities in the image, (iii) visual reasoning using different views generated from the segmentations. They further propose the inclusion of a coordinating agent for planning and review. They report in-domain and out-of-domain improvements over comparable methods and include a comprehensive ablation study evaluating various components of their method.

**Strengths:**

The paper proposes a novel framework for agentic visual reasoning with tool usage in medicine. They identify core workflow steps helpful for medical visual reasoning and train dedicated agents for these tasks. While the core concept of tool-supported visual reasoning exists for generalist VQA with similar training strategies, the originality and significance of this work arises from a well-executed specialization for the medical domain that I can see to be easily built upon.
The method, modules and training strategy is well motivated and thoroughly evaluated and ablated.
The manuscript is well structured and explanations and figures are clear.

**Weaknesses:**

1. ##### **Missing related work:**

   The paper claims to beat state-of-the-art on VQA-RAD and SLAKE, however, there are multiple methods with better performance on VQA-RAD and SLAKE not mentioned in the comparison, e.g. \[1,2,3\]. These works should be included for a complete contextualization of the work and the claim adapted.

2. **Unclear contribution of the coordinator**

	In figure 3 and 18 we can see that the coordinator (GPT-5) is able to overwrite the answers of the medical entity proposal module as well as the final answer of the system. Especially on the OOD dataset, where GPT-5 alone outperforms CARE, I am wondering how often the coordinator makes use of this power, essentially circumventing the contributions of all the proposed modules and degenerating to an API call to ChatGPT (albeit with the input of the CARE modules). A different reading of the OOD results would be that GPT-5 reaches an accuracy of 62.20 but if we include all the info from the CARE pipeline, GPT-5 becomes worse.

\[1\] Yu, Ting, et al. "Fine-grained Adaptive Visual Prompt for Generative Medical Visual Question Answering." *Proceedings of the AAAI Conference on Artificial Intelligence*. Vol. 39\. No. 9\. 2025\.
\[2\] Cui, Hejie, et al. "Biomedical visual instruction tuning with clinician preference alignment." *Advances in neural information processing systems* 37 (2024): 96449-96467.
\[3\] Lin, Weixiong, et al. "Pmc-clip: Contrastive language-image pre-training using biomedical documents." *International Conference on Medical Image Computing and Computer-Assisted Intervention*. Cham: Springer Nature Switzerland, 2023\.

**Questions:**

I would like to see some statistics on how often the final answer is overwritten by GPT-5 and the accuracy of these overwrites.

What is the binary modality token?

---

> ### Author Response · Authors · 2025-11-22
> **Reply to Reviewer fS8R**
>
> We thank the reviewer for the constructive and careful review, recognizing the novelty of the proposal framework that originated from medical domain specialization, detailed empirical evaluation, and a well-structured manuscript.
>
> 1. **Missing related work.**
>
>     We thank the reviewer for these valuable references, and we have already included them in the revised paper's discussion in Sec. 5. However, we highlight that their training data, settings, and metrics differ significantly from ours.
>     - FAVP [1] focuses on adaptive visual prompting, an orthogonal task. They used a much larger, 3-stage fine-tuning process involving $\sim$227k PMC-VQA data, compared to our 10k training pairs. Their reported closed-end accuracy and open-end recall are not directly comparable to our overall accuracy (LLM-as-judge). Comparing only closed-end accuracy: our method achieves **78.88%** on VQA-RAD and **89.19%** on SLAKE (which is higher than FAVP's **88.1%**).
>     - BioMed-VITAL [2] is similarly pre-trained with a larger VQA dataset (derived from PMC-15M) and also reports closed-end accuracy and open-ended recall. Our results on closed-end questions in the SLAKE dataset are relatively comparable to this model (**89.19%** vs **90.70%**).
>     - PMC-CLIP [3] uses a completely different scheme: CLIP-style zero-shot classification with a limited candidate answer set for open-ended questions. Our method outperforms it on closed SLAKE evaluation (**89.19%** vs **87.82%**).
>
>     We have acknowledged these works in the revised paper, Sec. 5 (line 528). While these works achieve higher accuracy on VQA-RAD, they utilize ~20x more training data. Our method achieves comparable (or better) results on SLAKE and OMVQA with significantly less data, highlighting the efficiency of the agentic decomposition. We further emphasize the novelty and advantage of our method: providing better accountability through task decomposition and evidence-grounded reasoning, which is critical for reliable medical AI.
>
>     We are currently conducting a human evaluation on the reasoning quality of our method, which will be added no later than the camera-ready, as it takes time to collaborate with experts. This can help highlight our advantage in clinical accountability.
>
>
> 2. **Unclear contribution of the coordinator and answer overwritten ratio by GPT-5.**
>
>     We agree that reporting the coordinator error ratio and its successful overwrite rate is helpful for understanding its contribution and behavior. We report the following values for each dataset: **Wrong->Correct** (ratio of cases where the coordinator successfully fixed the expert model's wrong answer) and **Correct->Wrong** (ratio where the coordinator mistakenly overwrites the expert model's correct answer). The difference between these two values is reported as the coordinator's positive contribution, and the sum of them represents the ratio of meaningful edits from the coordinator.
>
>                                                   Table A. Coordinator Edit Evaluation
>
>     | Dataset | Wrong→Correct | Correct→Wrong | Δ﻿ with Coordinator | Total Overwrite Rate |
>     | --- | :---: | :---: | :---: | :---: |
>     | OMVQA-3k | **1.90%** | 0.57% | +1.33% | 2.47% |
>     | VQA-RAD | **7.09%** | 4.87% | +2.22% | 11.96% |
>     | SLAKE | 2.77% | **3.15%** | -0.38% | 5.92% |
>     | VQA-Med-2019 (OOD) | **7.60%** | 3.60% | +4.20% | 11.20% |
>     | Overall | **4.84%** | 3.05% | +1.79% | 7.89% |
>
>     We note that the overall coordinator overwriting ratio is less than 12%, meaning most final answers still come from the expert VLM. However, we agree that smaller VLMs' reasoning traces are more error-prone on unseen data, leading to a larger coordinator contribution in OOD cases. We clearly instructed the coordinator (Figure 10) to only review CoT and answer alignment, not to make its own decision. The coordinator serves only as a verifier and tool-invocation planner, **leveraging GPT-5's strong reasoning capability to improve the VLMs' output, not its internal knowledge.** As shown by the in-domain dataset performance in Table 1, CARE-Coord outperforms the GPT-5 baseline (83.09% vs 68.56%), which further demonstrates that the coordinator is not playing the decisive role in the final answer.
>
>     This result is added to the main paper in the Sec. 4.2 and Table 8, starting from line 471.
>
> 3. **What are binary modality tokens?**
> The binary modality token $t_{mod}$ in Section 3.2 is a binary sequence matching the length of the concatenated image and text tokens. Inspired by positional encodings, $t_{mod}$ is set to 0 for image tokens and 1 for text tokens. This helps the SAM encoder layers distinguish between the two modalities. We have updated the description for our binary modality token in the revised paper (line 212).

---

> > ### Comment · Reviewer_fS8R · 2025-11-24
> >
> > 1. **Missing related work**:
> >
> > I appreciate the authors reply and the extension of the limitations section, however, I still think these methods should be included in the main comparison and your claims should be adapted. I follow the authors that open-end evaluation of PMC-CLIP [3] is done in a different setting due to the use of answer candidates, however closed evaluation for all the mentioned methods [1, 2, 3] and open-end evaluation on [1, 2] should be compared against, as these methods are performing the same task (VQA) on the same datasets (VQA-RAD, SLAKE) with the same inputs. I find the argument that the other methods are using more training data to be slightly inconsistent as you are numerically comparing to many other baselines which use even more training data. I understand that your evaluation differs from these other methods but I don’t think this is a reason not to compare at all, as it should not be difficult to evaluate your method with closed accuracy and open-end recall.
> >
> > To be clear, I agree that your method offers novelty and advantages over these other methods and I would raise my score even when your performance is worse than SOTA. However, I would not recommend acceptance if this performance comparison is not transparent and clear and your claims about achieving new SOTA (examples below) are not adapted.
> >
> > In the paper you claim in the Introduction:
> > “CARE-Flow (totaling 10B parameters) achieves new state-of-the-art (SOTA) performance, outperforming substantially larger public models, demonstrating strong parameter efficiency.”
> >
> > And in the Conclusion:
> > “Using a robust coordinator model like GPT-5 further expanded the capability of CARE, achieving state-of-the-art with a smaller model size.”
> >
> > 2. **Unclear contribution of the coordinator and answer overwritten ratio by GPT-5**
> >
> > Thank you for adding this analysis. It is very helpful to understand the coordinator contribution.
> >
> > Minor point: There is a typo in your updated manuscript in the caption of table 8.

---

> > > ### Author Response · Authors · 2025-11-26
> > > **Reply to Reviewer fS8R**
> > >
> > > We sincerely thank the reviewer for the prompt feedback and the willingness to re-evaluate our score. We absolutely agree with your emphasis on transparency and precise scientific claims. We have updated the manuscript to address your concerns:
> > >
> > > 1. **Comparison with Related Works [1, 2, 3]**
> > >
> > >     | Method | VQA-RAD (Open Recall) | VQA-RAD (Closed Accuracy) | SLAKE (Open Recall) | SLAKE (Closed Accuracy) |
> > >     | --- | :---: | :---: | :---: | :---: |
> > >     | FAVP - Vicuna | **71.90** | **88.20** | 87.20 | 88.10 |
> > >     | BioMed-VITAL-13B | 69.72 | 84.86 | **91.69** | **90.70** |
> > >     | PMC-CLIP | - | 84.00 | - | 88.00 |
> > >     | Ours (CARE-Coord-B) | 66.27 | 78.88 | 87.34 | 89.19 |
> > >
> > >     We have added a dedicated comparison with FAVP [1], BioMed-VITAL [2], and PMC-CLIP [3] in the **main paper Sec. 4.2 and Table 9**.
> > >
> > >     We placed this in the main paper to ensure a mathematically valid comparison, as these baselines report **Recall** for open-ended questions, whereas our main table uses **Accuracy (LLM-as-Judge)**. Merging these distinct metrics into one table would be misleading. Unfortunately, we were unable to reproduce these baselines using evaluation settings and metrics because [1] and [2] lack runnable training/inference codes, training environment settings, and pre-trained checkpoint (FAVP[1] only provides checkpoints for VQA-RAD, while BioMed-VITAL didn’t provide the environment configuration and evaluation code) for full reproduction, and [3] is a classification-based method incompatible with open-ended generation.
> > >     However, to ensure full transparency as requested, **we evaluated our CARE model using the specific metrics reported by these papers** (Open-Ended Recall and Closed-Ended Accuracy). This allows for a direct comparison in the new Table 9. We acknowledge that while our method is highly efficient and competitive on SLAKE Closed-Ended Accuracy (89.19%), these specialized baselines achieve higher scores on VQA-RAD.
> > >
> > > 2. **Modification of SOTA claims**
> > >
> > >     We agree that claiming "SOTA" broadly is inaccurate given the performance of the specialist models mentioned. We have revised the claims in the Introduction and Conclusion to be precise.
> > >
> > >     - **Original**: “CARE-Flow (totaling 10B parameters) achieves new state-of-the-art (SOTA) performance, outperforming substantially larger public models, demonstrating strong parameter efficiency.”
> > >     - **Modified**: “CARE-Flow (totaling 10B parameters) shows strong competitive results on multiple benchmarks, outperforming comparable generalist models and demonstrating strong parameter efficiency.” (in line 95)
> > >     - **Original**: “Using a robust coordinator model like GPT-5 further expanded the capability of CARE, achieving state-of-the-art with a smaller model size.”
> > >     - **Modified**: “Using a robust coordinator model like GPT-5 further expanded the capability of CARE, demonstrating competitive accuracy in both ID and OOD settings.” (in line 535).
> > >
> > >     This helps ensure a precise and professional expression in terms of model performance.
> > >
> > > 3. **Typo Correction.**
> > >
> > >     Thank you for noting this. We have corrected the typo in the caption of Table 8.

---

> > > > ### Author Response · Authors · 2025-11-26
> > > > **Additional Human Evaluation**
> > > >
> > > > Additionally, as suggested by the reviewer **G9oY** and as promised before, we here post an initial result for the human evaluation on the reasoning trace of our model.
> > > >
> > > >                     Table C. Additional Human Evaluation on Reasoning Trace
> > > >
> > > > | **Coordinator** | Human Evaluation Pass Rate (%) |
> > > > | --- | :---: |
> > > > | GPT-4o | 73.94 |
> > > > | GPT-5 (CARE-Coord-B)  | 82.14 |
> > > >
> > > > - **User Study Setting**: We randomly sample 35 correctly answered cases from the four test datasets to evaluate the accountability of the reasoning traces.  We develop a web-based evaluation platform, where human evaluators can examine the full reasoning process and assign a True/False judgment for each case (the user study interface is shown in Figure 24).  We recruit nine medical students to perform the evaluations through the platform. For comparison, we also included GPT-4o as the coordinator baseline.
> > > > - **Experiment Participant**: Due to the limited time, we are only able to contact participants with PhD/MD level knowledge for our experiments. We gathered 9 feedback from 9 participants with no knowledge about our work, all of them either have obtained or are pursuing a PhD/MD degree related to the medical and imaging domain. We plan to further collaborate with experts with clinical experience later, and we agree that this is a critical step towards real-world application.
> > > > - **Results**: We report the results in the Table above (Table 18 in the paper). CARE-Coord-B achieved a **human evaluation pass rate of 82.14%**, surpassing the GPT-4o baseline (73.94%). This result demonstrates that our proposed framework not only achieves higher accuracy but, more importantly, generates reasoning traces that are more factually accurate and visually grounded, thereby offering superior clinical accountability.
> > > >
> > > > This evaluation is now added to the revised paper in Appendix C.7 and Table 18.

---

> > > > > ### Comment · Reviewer_fS8R · 2025-11-26
> > > > >
> > > > > Thank you for adding the comparison and adapting the claims. This helps contextualize the work within the literature. All my concerns are now addressed.
> > > > > As I said earlier, I believe that even though these methods achieve better performance, the interpretability, generality and efficiency of CARE offers great advantages that make this work a valuable contribution to ICLR. I increase my score to reflect this.

---

> > > > > > ### Author Response · Authors · 2025-11-26
> > > > > >
> > > > > > We sincerely appreciate your reply and inspiring suggestion throughout the discussion, particularly regarding the baselines' transparency and precision of our claim. Your comment really helps strengthen our manuscript in terms of professionalism and clarity of presentation. We are delighted that you recognize the value of CARE in terms of interpretability, generality, and efficiency. We will insist on these improvements and incorporate them into our final version of the paper as it is.
> > > > > >
> > > > > > Once again, we thank you very much for your time and invaluable help during the discussion period.

---

### Official Review · Reviewer_ojM1 · 2025-10-28

**Soundness:** 3
**Presentation:** 4
**Contribution:** 3
**Rating:** 6
**Confidence:** 3

**Summary:**

This paper proposes CARE, an agentic framework that mirrors the clinical diagnostic workflow. Instead of treating medical visual question answering (VQA) as a monolithic black-box task, the authors decompose it into interpretable sub-steps: (1) a medical entity proposal stage, (2) a segmentation model to localize regions of interest (ROIs), and (3) an evidence-grounded VQA model that reasons over the full image along with different evidence views (zoom-in, masked, or global). Additionally, an LLM-based coordinator performs planning, reasoning, and quality control over these steps, improving performance and interpretability.

Overall, I find this to be a promising and well-motivated direction. The framework reflects clinical reasoning in a meaningful way, and the results appear solid across multiple datasets.

**Strengths:**

- Well-motivated workflow design. The authors tackle the core problem—controlling hallucinations in large medical VLMs—through a structured, interpretable workflow rather than pure end-to-end training. This decomposition aligns well with clinical reasoning and is conceptually elegant.

- Coordinator design. The introduction of an LLM-based coordinator for planning and quality control is intuitive and methodologically sound. It also adds an additional layer of reliability that most prior medical VQA systems lack.

- Strong empirical performance. The reported results show consistent improvement over prior works, demonstrating the practical value of the agentic workflow.

- Extensive experimentation. The authors conduct a large number of experiments, including multiple baselines and ablations, which adds credibility to the study.

**Weaknesses:**

See questions

**Questions:**

- Reproducibility and openness. Since the paper claims superior performance, open-sourcing the models, weights, and codebase would be critical for community validation. Without this, it’s difficult to verify the robustness and generality of the framework.

- Justification for reinforcement learning. The ablation study supports the inclusion of RL fine-tuning, but the paper should better articulate why reinforcement learning is necessary for both the entity proposal and the VQA stages. From the description, it appears to combine symbolic and correctness-based rewards; please clarify why supervised fine-tuning (SFT) would not suffice.

- Demonstration of diagnostic effectiveness. If the paper aims to position CARE as a clinically meaningful diagnostic model, the current visual examples (e.g., Figure 3) feel too trivial. The task shown (“Which organ has the largest area?”) is not representative of real diagnostic challenges and could likely be handled by standard detection models (e.g., YOLO). Consider adding more realistic examples—such as differentiating between cancerous vs. non-cancerous lung images—and illustrating how the coordinator or CoT reasoning contributes to that decision.

---

> ### Author Response · Authors · 2025-11-22
> **Reply to Reviewer ojM1**
>
> We thank the reviewer ojM1 for the inspiring and helpful comments. We are glad to see that the reviewer recognized our well-motivated workflow design with a coordinator planning, and our strong empirical performance from an extensive amount of experiments. We address the concerns and questions about our work.
>
> 1. **Reproducibility and Openness.**
>
>     We completely agree that **openness is key** to a healthy research community. As stated in the Reproducibility Statement (Appendix A), we plan to **release the full training and inference code, data split, synthetic data, and pre-trained model weights upon publication**. Furthermore, as reviewer G9oY noted, we already included a very detailed implementation recipe in the paper, covering model architecture, hyperparameters, model prompts, and tool schema definition, which aids in both understanding and reproduction.
>
>
> 2. **Justification for reinforcement learning. Please clarify why supervised fine-tuning (SFT) would not suffice.**
>
>     We clarify that we apply **SFT+RFT only on the EG-VQA model**, but **only RFT on synthetic data for the entity proposal model**.
>
>     Recent studies [1, 2] suggest SFT injects new knowledge (memorization), while RFT improves existing capabilities by adjusting output to generate a reasonable Chain-of-Thought (CoT). Our training reflects this: (1) The **entity-proposal model** (InternVL3) is already pre-trained on medical data (VQA and image captioning), so we use RFT to refine its capability and provide a convincing CoT. (2) The **EG-VQA model** faces a new diagnosis task (input + visual evidence) that needs to make use of additional visual evidence. Since this is outside its base capability, we first use SFT to teach the task, followed by RFT to improve reasoning and accuracy. Overall, we use RFT for both to ensure strong reasoning and clinical accountability under limited data.
>
>     Table 3 validates this. For EG-VQA, combining SFT and RFT (**SFT+RFT/DAPO**) yielded the largest improvement (**+9.6%**), compared to a maximum of 7.2% for either method alone. The fact that SFT-only performs better than RFT-only implies that **pure RFT is insufficient** when the EG-VQA task is beyond the base model's initial knowledge.
>
>     This discussion is added to the revised paper in Sec. 3.4, starting from line 259.
>
>     Moreover, we report the performance of the entity proposal VLM with supervised fine-tuning (SFT) and SFT + RFT to validate our design. We use the static CARE-Flow framework here to isolate the influence of the coordinator review.
>
>               Table A. Ablation of Entity Proposal VLM Fine-tuning
>
>     | Fine-tuning | Entity Accuracy | Mask Dice | Overall MedVQA |
>     | --- | :---: | :---: | :---: |
>     | SFT | 76.70 | 72.69 | 74.26 |
>     | SFT+RFT | 80.92 | 74.03 | 74.44 |
>     | RFT (CARE-Flow-B) | **85.28** | **73.48** | **74.91** |
>
>     As shown above, using SFT alone or combining SFT and RFT underperforms our RFT model, which validates our choice of training recipe. This suggests that applying SFT with limited synthetic data may be harmful to our task. Applying RFT over SFT indeed helps general performance, but for a task that is already within the model’s capability, directly applying RFT could be the best option.
>
>     This experiment is added to the revised paper in Appendix C.6 and Table 16, starting from line 1182.
>
>     **References**:
>
>     [1] Chu, Tianzhe, et al. "Sft memorizes, rl generalizes: A comparative study of foundation model post-training." *arXiv preprint arXiv:2501.17161* (2025).
>     [2] Ma, Lu, et al. "Learning What Reinforcement Learning Can't: Interleaved Online Fine-Tuning for Hardest Questions." *arXiv preprint arXiv:2506.07527* (2025).
>
> 3. **Demonstration of diagnostic effectiveness on more realistic examples.**
>
>     We appreciate the suggestion about the case study, and we have replaced the example in Figure 3 with a more complex disease identification question that demands expert reasoning. Meanwhile, we move the original example to Figure 19, and we have provided more clinically-centered examples in the Appendix C.11, including Figures 14-18 (covering brain tumor localization, anomaly diagnosis for both healthy and unhealthy cases, skin lesion detection, COVID-19 diagnosis, and others), as well as pulmonary infiltration (Figure 1) and lung cancer (Figure 2) cases. These examples demonstrate the reasoning path and the capability of our framework in more complex, knowledge-demanding clinical and diagnostic situations.
>
>     We have highlighted those examples in the revised paper in line 405.

---

### Official Review · Reviewer_G9oY · 2025-10-29

**Soundness:** 3
**Presentation:** 3
**Contribution:** 3
**Rating:** 6
**Confidence:** 3

**Summary:**

The paper introduces CARE, an agentic framework for medical VQA that explicitly mirrors clinical workflows. CARE decomposes the task:
(i) a medical entity proposal VLM (trained with RLVR/DAPO using verifiable rewards), (ii) an entity-referring segmentation module (SA-Med-2D) providing pixel-level evidence (ROI masks/boxes), and (iii) an evidence-grounded VQA (EG-VQA) reasoner that answers using one of three evidence views: Zoom-in, Mask, or Global (a full-context “no localization needed” indicator).

Two operating modes are studied: CARE-Flow (deterministic pipeline) and CARE-Coord (a planner–reviewer agent that selects tools, iterates, and performs answer reviews). Across four benchmarks, the 10B CARE-Flow establishes strong new results; CARE-Coord further improves performance, surpassing prior 32B baselines. However, it's heavily dependent on using relatively bigger proprietary models such as GPT-5.

**Strengths:**

The paper's primary strength lies in its principled and clinically inspired architecture. The decomposition of medical reasoning into discrete, specialist-driven stages [hypothesize (entity proposal) → localize (segmentation) → reason (grounded VQA)] is a significant conceptual advance. This design directly addresses the "black box" problem by providing explicit, pixel-level evidence for each decision, thereby enhancing both interpretability and debuggability. Empirically, the work is robust, demonstrating strong performance across multiple datasets and supporting its claims with thorough ablation studies that dissect the contribution of each component, from the matching strategy to the agentic coordinator. The use of reinforcement learning with verifiable rewards (RLVR/DAPO) is a sophisticated technical choice that aligns model incentives with checkable signals, avoiding the need for costly human preference data. Finally, the paper is commendably detailed in its implementation description, providing prompts and tool schemas that greatly exceed the reproducibility standards of many contemporary agentic AI papers.

**Weaknesses:**

### **Baseline Comparison Fairness and Transparency**
The validity of the primary results in Table 1 depends critically on understanding each baseline’s training regime. It remains unclear which comparison models were fine-tuned on the same in-domain datasets (OmniMedVQA, VQA-RAD, SLAKE) and which were evaluated zero-shot. It is indeed unjust to call the columns in-domain and OOD solely based on your configurations and then compare them against other models. The authors should explicitly disclose the training exposure for every baseline—ideally in a small supplementary table—and group results into “Fine-tuned” vs. “Zero-shot/Unknown” categories. Without this disclosure, the magnitude of CARE’s improvement cannot be properly attributed to architectural design versus training-data advantage. The current Table also needs to be modified or at least clearly states the differences and point to the more complete Table which would be in the appendix.

### **Reward Function Analysis and Sensitivity**
While the EG-VQA reward is ablated, the Entity Proposal reward R_{\text{Entity}} = R_{\text{sim}} + R_{\text{count}} + R_{\text{repetition}} + R_{\text{format}} lacks empirical justification for its composition.
A component-wise ablation analysis or a reference to another paper that is using the same reward with clear ablations would clarify which sub-rewards most influence stability and downstream performance. This is particularly important since R_{\text{Entity}} governs the quality of the entity grounding that underpins the rest of the pipeline.

### **Isolating Framework Contributions (CARE-Coord-Open Variant)**
The performance of CARE-Coord depends in part on its proprietary coordinator (GPT-5). To distinguish architectural benefit from raw model capacity, the paper should include a CARE-Coord-Open variant using the strongest open-source VLM employed elsewhere (e.g., InternVL3-8B). Reporting this gap would isolate how much of the gain stems from the agentic framework itself versus the closed-model advantage.

### **Misleading Parameter-Scaling Visualization**
Figure 1(e) plots proprietary models with unknown parameter counts on a “Billions of Parameters” axis, inadvertently implying knowledge of their scale and suggesting a false comparison. The authors should redesign the plot; for instance, use a separate panel, shaded region, or qualitative marker to avoid misrepresenting scaling relationships.

**Questions:**

Also refer to **Weaknesses**.

### **Baseline Training Exposure and Evaluation Fairness**
For each baseline reported in Table 1, please explicitly indicate whether the model was:
(a) fine-tuned on any of the in-domain datasets (OmniMedVQA, VQA-RAD, SLAKE),
(b) fine-tuned on other medical VQA datasets, or (c) evaluated zero-shot using public checkpoints. This transparency is critical to disentangle architectural advantages from training-data advantages. A compact supplementary table summarizing this information would substantially improve interpretability and fairness.

### **Quantitative Evaluation of Accountability**
Beyond qualitative case studies, have you considered measuring accountability quantitatively (e.g., the overlap between predicted answer entities and the segmented ROI, or expert ratings of reasoning trace correctness)? Even a limited-scale human evaluation or automatic alignment metric would strengthen the accountability claim.

### **Coordinator Error and Review Conservatism**
In Figure 18, the coordinator overrides a correct specialist answer, illustrating a potential hallucination at the review stage.
How frequently do such coordinator-induced reversals occur across the evaluation set? Have you explored conservative review heuristics (e.g., vetoing coordinator changes when specialist confidence is high) to mitigate these errors?

### **Evaluation Stability and LLM-Based Scoring**
For open-ended questions evaluated by GPT-4-class models, have you measured inter-rater agreement (e.g., multiple LLM or human graders) or compared results with deterministic metrics like normalized exact match? Providing a measure of scoring variance would improve confidence in reported accuracy differences.

---

> ### Author Response · Authors · 2025-11-22
> **Reply to Reviewer G9oY (1/4)**
>
> We thank the reviewer for the detailed and constructive comments, and for recognizing our clinically inspired architecture, empirical performance improvement, and thorough ablation studies, and detailed implementation descriptions. We address the concerns and questions mentioned in the review point-by-point.
>
> 1. **Baseline Comparison Fairness and Transparency.**
>
>     We agree that transparency on baseline training data is crucial. The notion of In-domain (ID) and out-of-domain (OOD) in our paper mainly refers to the training setting of our own. For clarity, we provide a detailed summary of the medical training datasets for all baselines in the table below (Table A), highlighting those that overlap with ours.
>
>                                                 Table A: Training dataset comparison
>
>     | Model | Medical Data Training | Training Medical Datasets |
>     | --- | :---: | --- |
>     | GPT-4o / GPT-5 | ✓ | Unknown |
>     | Llama-3.2-11B-Vision | ✗ | N/A |
>     | Qwen2.5-VL | ✗ | N/A |
>     | InternVL3 | ✓ | **VQA-RAD**, **SLAKE**, and others |
>     | DeepEyes | ✗ | N/A |
>     | Llava-Med-v1.5-mistral-7b | ✓ | PMC-15M |
>     | MedVLM-R1-2B | ✓ | **OmniMedVQA**, **VQA-RAD,** **SLAKE**, and others |
>     | medgemma | ✓ | **VQA-RAD**, **SLAKE**, MIMIC-CXR, and others |
>     | HuatuoGPT-Vision | ✓ | PubMedVision |
>     | Lingshu | ✓ | **VQA-RAD**, **SLAKE**, **VQA-Med-2019**, and others |
>     | CARE (Ours) | ✓ | **OmniMedVQA**, **VQA-RAD**, **SLAKE** |
>
>     Most medical expert VLMs we report on were pre-trained or fine-tuned with overlapping medical data, often with significantly larger datasets (e.g., HuatuoGPT-Vision used over 1M data, Lingshu used over 12M). In contrast, our total VQA training data size is only just over 10k. While general VLMs like the Qwen series are not medically pre-trained, some other baselines (proprietary GPT family and InternVL3) did include medical data in their training. This table is added to Appendix C.1 and Table 9 in the revised paper.
>
>     We introduce InternVL3-Finetuned, a new InternVL3 baseline fine-tuned with our exact training data configuration to isolate the training dataset's influence. We report its performance here. We compare this new baseline with our static CARE-Flow model to isolate the influence of the external coordinator. Note that CARE-Flow-S uses the InternVL3-2B architecture for entity proposal and EG-VQA, while CARE-Flow-B uses InternVL3-2B for entity proposal and InternVL3-8B for EG-VQA.
>
>                                             Table B. Fine-tuned Baseline Comparison
>
>     | Model | OMVQA-3k | VQA-RAD | SLAKE | VQA-Med-2019 | Overall |
>     | --- | :---: | :---: | :---: | :---: | :---: |
>     | InternVL3-2B (zero-shot) | 75.43 | 55.65 | 63.07 | 54.00 | 62.04 |
>     | InternVL3-2B-Finetuned | 87.97 | 57.43 | 69.56 | 51.00 | 66.49 |
>     | CARE-Flow-S  | 94.53 | 56.32 | 78.44 | 53.60 | 70.72 |
>     | InternVL3-8B (zero-shot) | 75.97 | 61.86 | 66.13 | **57.40** | 65.34 |
>     | InternVL3-8B-Finetuned | 91.13 | 61.86 | 76.53 | 53.80 | 70.83 |
>     | CARE-Flow-B  | **96.17** | **63.64** | **83.21** | 56.60 | **74.91** |
>
>
>
>     The results show that, even with the same training and fine-tuning settings, our method consistently outperforms the baseline using the same base model by an average of over 4%. This highlights the contribution of our decomposed and evidence-grounded VQA pipeline.
>
>     This evaluation is added to the revised paper in Appendix C.6 and Table 13, starting from line 1129.

---

> ### Author Response · Authors · 2025-11-22
> **Replay to Reviewer G9oY (2/4)**
>
> 2. **Reward Function Analysis and Sensitivity.**
>
>     The Reward for Entity Proposal Model (RFT) is comprised of similarity reward ($R_{sim}$), count reward ($R_{count}$), repetition reward ($R_{repetition}$), and format reward ($R_{format}$). We previously showed that our design of soft $R_{sim}$ significantly outperforms a common binary accuracy reward (Table 6), yielding $>10\%$ improvement in entity proposal accuracy and $>20\%$ improvement in segmentation dice score. The count reward and repetition reward are designed, like $R_{format}$, to improve output format; $R_{count}$ limits the number of generated entities, and $R_{repetition}$ prevents duplicate proposals.
>
>     We provide more ablation experiments on this reward design for the entity-proposal VLM using the static CARE-Flow-B (10B) framework to isolate the external coordinator’s influence. $R_{format}$ is not ablated because it is the necessary default reward, as described in the original GRPO paper, used to control the output format.
>
>                         Table C. Entity Proposal Reward Ablation
>
>     | $$R_{count}$$ | $$R_{repetition}$$ | Entity Accuracy | Mask Dice | Overall MedVQA |
>     | :---: | :---: | :---: | :---: | :---: |
>     |  |  | 85.0 | 72.7 | 74.5 |
>     | ✓ |  | 84.7 | 72.7 | 74.3 |
>     |  | ✓ | 82.7 | 72.9 | 74.4 |
>     | ✓ | ✓ | **85.2** | **73.4** | **74.9** |
>
>     The additional format rewards, $R_{count}$ and $R_{repetition}$, help the model generate better formatted entity proposals, avoiding errors like repeated or excessive output. While this aids downstream evaluation and final VQA, their influence is relatively small since they only control the output format.
>
>     These results is added to the revised paper in Table 7 and Sec. 4.2, starting from line 464.
>
>
> 3. **Isolating Framework Contributions (CARE-Coord-Open Variant).**
>
>     We agree that it is essential to assess the impact and contribution of the external proprietary coordinator, particularly given that the proprietary model is not always available. In fact, we have trained and evaluated such an open variant in Table 4, and described it in detail in Appendix Sec. C.3. We have also evaluated the performance of zero-shot open VLM as a coordinator in Table 4. We present the evaluation of different coordinators below.
>
>                         Table D. Ablation on Different Coordinator
>
>     | Coordiantor | ID | OOD  | Overall |
>     | --- | :---: | :---: | :---: |
>     | Random choice  | 79.34 | 52.00 | 72.50 |
>     | Majority Vote (CARE-Flow, *need to enumerate the full tool set*)  | 81.01 | 56.60 | 74.91 |
>     | InternVL3-38B | 79.73 | 56.80 | 74.00 |
>     | InternVL3-8B + RFT (CARE-Coord-Open) | 80.74 | 58.20 | 75.10 |
>     | GPT-4o | 79.65 | 54.20 | 73.29 |
>     | GPT-5 (CARE-Coord) | **83.12** | **60.80** | **77.54** |
>
>     Our evaluation confirms that using a fine-tuned open-source coordinator improves overall medical VQA performance, particularly on the OOD dataset. Even a small, fine-tuned open coordinator contributes to the overall performance within our framework. While the SoTA proprietary coordinator (GPT-5) still outperforms the open-sourced version, this is expected given the size difference.
>
>     We have emphasized our open-sourced Coordinator in the revised paper, Sec. 4.2, starting from line 431.
>
>
> 4. **Misleading Parameter-Scaling Visualization**
>
>     We agree that plotting proprietary models with unknown parameter counts on a “Billions of Parameters” axis in Figure 1(e) could be misinterpreted. In the revision, we have redesigned Figure 1(e) in page 2 following the suggestions to avoid this misunderstanding. We further note that we have no knowledge about the size of these proprietary models.

---

> ### Author Response · Authors · 2025-11-22
> **Reply to Reviewer G9oY (3/4)**
>
> 5. **Quantitative Evaluation of Accountability.**
>
>     We agree that a quantitative evaluation of the accountability is critical to our proposed method.
>
>     We have already demonstrated accountability in the paper (Table 5 and 6) with the following metrics:
>
>     1. We reported the **entity proposal accuracy** (85.2%) and **segmentation mask Dice score** (73.4%) in Table 6. This performance is satisfactory, as our goal is providing visual evidence for grounded VQA, not pixel-level accurate segmentation.
>     2. We also compared our segmentation model against various baselines in Table 5, where our method showed the best performance.
>
>     Since Entity Accuracy measures if the correct anatomy was identified, and Dice measures if that anatomy was segmented correctly, the combination of these two metrics effectively proxies the alignment between reasoning and visual evidence, which naturally leading to a better accountability and visual-grounding.
>
>     We have initiated a study with clinical experts to evaluate reasoning traces. While this takes time, we commit to including these human evaluation results no later than the camera-ready version of the paper to further validate the accountability claim.
>
> 6. **Coordinator Error and Review Conservatism**
>
>     While overall performance improved with the coordinator-reviewed answers (Table 1), we agree that the coordinator error report ratio and successful overwrite rate are helpful. We report the following values for each dataset: (1) **Wrong->Correct:** Ratio where the coordinator **successfully fixed** the expert model's wrong answer. (2) **Correct->Wrong:** Ratio where the coordinator **mistakenly overwrote** the expert model's correct answer.
>
>     The difference between these two values represents the positive contribution from the coordinator, while the sum is the total ratio of meaningful coordinator editing.
>
>                                    Table E. Coordinator Edit Evaluation
>
>     | Dataset | Wrong→Correct | Correct→Wrong | $\Delta$ with Coordinator | Total Overwrite Rate |
>     | --- | :---: | :---: | :---: | :---: |
>     | OMVQA-3k | **1.90%** | 0.57% | +1.33% | 2.47% |
>     | VQA-RAD | **7.09%** | 4.87% | +2.22% | 11.96% |
>     | SLAKE | 2.77% | **3.15%** | -0.38% | 5.92% |
>     | VQA-Med-2019 (OOD) | **7.60%** | 3.60% | +4.20% | 11.20% |
>     | Overall | **4.84%** | 3.05% | +1.79% | 7.89% |
>
>     Overall, the coordinator overwriting ratio is **less than 12% in most cases**. While coordinator review can introduce errors, it generally performs better, likely due to its **stronger reasoning capability**. The coordinator is instructed only to **review the CoT from the VLMs**, making it a **verifier** based on provided information, not a final answer provider.
>
>     The coordinator shows a **higher "Wrong->Correct" rate in the OOD data**, demonstrating its **stronger generalization capability** where smaller VLMs often fail. Given its statistically better performance, it remains reasonable to trust the coordinator's results, concluding that our coordinator review design is helpful to the final performance.
>
>     This result is added to the main paper in the Sec. 4.2 and Table 8, starting from line 471.
>
>     We also explored a **conservative coordinator strategy** as suggested. Since the reasoning VLM lacks direct confidence, we asked it to output a **confidence score (0–100)** based on its reasoning. We apply a **hard threshold $\sigma$**: if the local VLM's confidence is $\geq\sigma$, we use the expert model's answer; otherwise, we use the coordinator's final answer. We report this final performance alongside our CARE-Coord-B in the table below.
>
>                                    Table F. Conservative Coordinator Strategy
>
>     | Model | OMVQA-3k | VQA-RAD | SLAKE | VQA-Med-2019 | Overall |
>     | --- | :---: | :---: | :---: | :---: | :---: |
>     | $\sigma=25$ | 96.70 | 66.29 | 83.77 | 56.40 | 75.79 |
>     | $\sigma=50$ | 96.74 | 67.40 | **83.87** | 56.60 | 76.15 |
>     |  $\sigma=75$ | 97.47 | 67.85 | 83.68 | 56.80 | 76.45 |
>     | CARE-Coord-B | **97.97** | **68.29** | 83.11 | **60.80** | **77.54** |
>
>     We observe that adapting the final answer based directly on the **expert VLM's confidence score** generally **does not improve performance**, consistent with previous evaluations. Our coordinator's CoT review process already **implicitly considers the confidence** of the expert VLM's reasoning trace, as we ask it to review the CoT quality. Consequently, the coordinator insists on the expert VLM's output when its reasoning trace is confident.
>
>     This result is added to the revised paper in Appendix C.6 and Table 14, starting from line 1165.
>
>     However, we acknowledge that asking the VLM to directly output its confidence may not be the most reasonable practice, and we will consider this further in future research.

---

> ### Author Response · Authors · 2025-11-22
> **Reply to Reviewer G9oY (4/4)**
>
> 7. **Evaluation Stability and LLM-Based Scoring.**
>
>     Following Lingshu, we use **GPT-4o as our LLM judger** for open-ended questions. We set the LLM-as-judge to **zero temperature**, which ensures **deterministic and relatively stable behavior** across different baselines, enabling a fair comparison, though we acknowledge potential mistakes.
>
>     We also evaluate our results with different LLM-as-judges (both proprietary and open-source) to demonstrate the **variance stemming from the judger**. We skip OmniMedVQA in this evaluation as it only contains closed-ended questions.
>
>                               Table G. Ablation of Different LLM-as-Judge
>
>     | LLM Judger | VQA-RAD | SLAKE | VQA-Med-2019 | Overall |
>     | --- | :---: | :---: | :---: | :---: |
>     | GPT-4o | 68.29 | 83.11  | 60.80 | 70.73 |
>     | GPT-4o-mini | 68.51 | 84.35 | 60.60 | 71.15 |
>     | InternVL3-38B-Instruct | 67.41 | 82.44 | 59.00 | 69.62 |
>     | InternVL3-78B-Instruct | 67.41 | 83.30 | 60.20 | 70.30 |
>     | Avg. | 67.91 | 83.30 | 60.15 | 70.45 |
>     | STD | ±0.58 | ±0.79 | ±0.81 | ±0.66 |
>
>     We note that the variance between different LLM-as-judges is very small (less than 1%). Our performance improvement is significant enough considering this variance.
>
>     This result is added to the revised paper in Appendix C.6 and Table 15, starting from line 1176.

---

> ### Author Response · Authors · 2025-11-26
> **Human Evaluation on Reasoning Trace**
>
> As suggested by the reviewer, we here post an initial result for the human evaluation on the reasoning trace of our model.
>
>                     Table H. Additional Human Evaluation on Reasoning Trace
>
> | **Coordinator** | Human Evaluation Pass Rate (%) |
> | --- | :---: |
> | GPT-4o | 73.94 |
> | GPT-5 (CARE-Coord-B)  | 82.14 |
>
> - **User Study Setting**: We randomly sample 35 correctly answered cases from the four test datasets to evaluate the accountability of the reasoning traces.  We develop a web-based evaluation platform, where human evaluators can examine the full reasoning process and assign a True/False judgment for each case (the user study interface is shown in Figure 24).  We recruit nine medical students to perform the evaluations through the platform. For comparison, we also included GPT-4o as the coordinator baseline.
> - **Experiment Participant**: Due to the limited time, we are only able to contact participants with PhD/MD level knowledge for our experiments. We gathered 9 feedback from 9 participants with no knowledge about our work, all of them either have obtained or are pursuing a PhD/MD degree related to the medical and imaging domain. We plan to further collaborate with experts with clinical experience later, and we agree that this is a critical step towards real-world application.
> - **Results**: We report the results in the Table above (Table 18 in the paper). CARE-Coord-B achieved a **human evaluation pass rate of 82.14%**, surpassing the GPT-4o baseline (73.94%). This result demonstrates that our proposed framework not only achieves higher accuracy but, more importantly, generates reasoning traces that are more factually accurate and visually grounded, thereby offering superior clinical accountability.
>
> This evaluation is now added to the revised paper in Appendix C.7 and Table 18.

---

> > ### Comment · Reviewer_G9oY · 2025-11-28
> >
> > Thank you for your thorough and detailed response. Most of my concerns are answered; therefore, I will raise my score to an "8."

---

> > > ### Author Response · Authors · 2025-11-28
> > >
> > > We sincerely appreciate the reviewer for the insightful, rigorous, and constructive guidance throughout the discussion and review period. Your comment has helped us improve our manuscript greatly regarding the baseline fairness and reasoning accountability evaluation. We commit that all these improvements and discussions will be reflected in the final version of the paper.
> > >
> > > Again, we thank you deeply for the reviewer's positive feedback nd the decision to raise the score.

---

### Author Response · Authors · 2025-11-22
**General Reply & Rebuttal Period Paper Revision Summary**

We sincerely appreciate all the reviewers for their valuable and insightful comments. We are glad to see that all the reviewers have praised our work for its well-motivated architecture, extensive experiments, and ablation studies. According to the comments, we have improved our work and addressed the concerns and questions for each reviewer individually.

We have also updated our paper accordingly and highlighted the changes in the paper with blue text, and highlighted existing content in light red. Below, we summarize all the changes:

For Reviewer **G9oY**:

1. (**G9oY**) We have added the information about the medical training dataset for each baseline in Appendix C.1 and Figure 9.
2. (**G9oY**) We have added a new evaluation comparing with a fine-tuned InternVL3 model with the same training data in Appendix C.6, paragraph of “Fine-tuned Baseline” and Table 14.
3. (**G9oY**) We have added a new ablation experiment about the entity proposal reward function in Sec. 4.2, paragraph of “Ablation of Entity Proposal Reward” and Table 7.
4. (**G9oY**) We have added a highlight about the open-source coordinator evaluation in the Sec. 4.2, paragraph of “Ablation on Coordinator” in lines 427 and 431.
5. (**G9oY**) In Figure 1, we have updated sub-figure (e) to avoid misleading proprietary model size.
6. (**G9oY**) We have highlighted our discussion on the entity proposal accuracy and segmentation performance in Sec. 4.2, paragraph of “Ablation on Entity Proposal VLM.” and Table 6.
7. (**G9oY**) We have added a new evaluation on the Coordinator’s edits in Sec. 4.2, paragraph “Evaluation of Coordinator Edits” and Table 8.
8. (**G9oY**) We have added a new evaluation about conservative coordinator strategy in Appendix C.6, paragraph of “Conservative Coordinator Strategy” and Table 15.
9. (**G9oY**) We have also evaluated with different LLM-as-Judge in Appendix C.6, paragraph of “LLM-as-Judge Stability” and Table 16.
10. (**G9oY**) We have added the human evaluation on the reasoning trace of our model in Appendix C.7 and Table 18.

For Reviewer **ojM1**:

1. (**ojM1**) We highlight the reproducibility statement in Appendix A about code release and openness.
2. (**ojM1**) We have added more discussion on why choosing RFT in Sec. 3.4.
3. (**ojM1**) We have added a comparison of different Entity Proposal VLM fine-tuning strategies (SFT vs RFT) in Appendix C.6, paragraph of “Fine-tuning Strategy for Entity Proposal Model” and Table 17.
4. (**ojM1**) We have replaced the original example in Figure 3 with a more complex clinical-related example for lung anomaly identification. The original example is moved to Figure 19.
5. (**ojM1**) We have added a highlight on the clinical-related examples in Sec. 4.1, pointing to more examples in Appendix C.11, Figure 14-18.

For Reviewer **fS8R**:

1. (**fS8R**) We have added an acknowledgement and more discussion about noted related works and their performance in the Sec. 5, Limitations and Future Work.
2. (**fS8R**) We have added a new evaluation on the Coordinator’s edits and discussion on the OOD performance in Sec. 4.2, paragraph “Evaluation of Coordinator Edits” and Table 8.
3. (**fS8R**) We have added an explanation about the design binary modality token in Sec. 3.2.
4. (**fS8R**) We have added a full comparison against the mentioned baselines [1,2,3] in Sec. 4.2 and Table 9. We have also modified our claim about SOTA in the introduction and conclusion, accordingly.
5. (**fS8R**) We have added the human evaluation on the reasoning trace of our model in Appendix C.7 and Table 18.

---

### Author Response · Authors · 2025-11-29
**Final Rebuttal Statement**

We sincerely appreciate the reviewers and Area Chairs for their contributions throughout the review and discussion period. We recognize that this is a very special moment for our community; however, we believe that your time and effort dedicated to maintaining the integrity and scholarly atmosphere of our field are all the more meaningful because of it. We acknowledge the additional effort required, particularly from our Area Chairs, to conclude the review process, and we are deeply grateful for your contribution.

To facilitate the subsequent decision-making process, we summarize the outcomes and discussions from the rebuttal period below:

First, we are glad to report that we have successfully addressed the key concerns raised by Reviewer **G9oY**. We provided seven new tables and corresponding analyses, offering additional clarification on training data fairness, human evaluation, coordinator error analysis, and all other concerns. These new evaluations make our claims more robust and convincing, and all related content has been incorporated into the revised manuscript. It is our great pleasure to note that Reviewer **G9oY** has agreed to **raise the score from 6 (WA) to 8 (A)**. We sincerely thank Reviewer **G9oY** for their detailed and insightful comments.

Second, we addressed the concerns and questions from Reviewer **ojM1**. We specifically highlighted the rationale behind our fine-tuning recipe and demonstrated the necessity of applying SFT and RFT to different tasks and models through additional explanations and experiments. As suggested, we also replaced the visual example in Figure 3 with a more representative case that requires clinical knowledge. Although the discussion period has concluded, preventing a direct reply, we believe our new clarifications and experiments adequately address these concerns and significantly improve the quality of our manuscript. We thank Reviewer **ojM1** for their **positive review (score of 6, WA)** and constructive suggestions, which we have incorporated into the revised paper.

Finally, we have resolved the concerns raised by Reviewer **fS8R** through a helpful and informative discussion. We fully agree with Reviewer **fS8R** regarding the transparency of performance comparisons; in conclusion, we have added comparisons with new SOTA baselines and refined our claims with more precise descriptions in the revised paper. These modifications ensure our paper makes professional and precise scientific statements, thereby improving the overall presentation of our work. As a result of our discussion, Reviewer **fS8R** has **raised the score from 6 (WA) to 8 (A)**. We once again thank Reviewer **fS8R** for the inspiring review and for recognizing our work’s advantages in accountability, generality, and efficiency.

Overall, we have had a highly productive discussion period; two of the three reviewers engaged with our rebuttal and eventually raised their scores. We believe this rebuttal discussion has been meaningful, significantly enhancing the quality of our paper through the reviewers' guidance. All of the reviewers have agreed on the novelty of our method and praised our extensive experiment and improved clinical accountability and performance. **Ultimately, we are happy to see that all reviewers maintain a positive outlook on our work and recommend Weak Acceptance or higher.**

Once again, we deeply thank all the reviewers and Area Chairs for their effort and invaluable service to this community. Your commitment and contributions are essential to maintaining the integrity and fairness of the process, particularly under such an uncommon situation.

---

### Meta-Review · Area_Chair_T5GH · 2026-01-02

**Summary:**

The reviewers' main concerns centered on a few main points. (1) Reliance on a proprietary model, which limits reproducibility and opens questions about the source of performance. In the rebuttal, the authors added requested comparisons and satisfied the reviewers. (2) Lack of clarity around source of performance (similar to (1)). In the rebuttal, the authors included a new experiment finetuning an alternative method. I suggest extending this experiment to report hyperparameter tuning settings to ensure there's no implementation bias here, where the proposed method is tuned with more care than a contender. Still, the added results helps boost the paper's claims.

**Reviewer Concerns:**

The reviewers' concerns were overall address sufficiently, though all improvements should be made to the paper prior to its publication.

**Reviewer Scores:**

Reviewers scores would likely increase based on the authors' responses.

---

### Decision · Program_Chairs · 2026-01-26

Accept (Poster)